# A data-driven method for estimating the composition of end-members from stream water chemistry timeseries

Esther Xu Fei[1] and Ciaran Joseph Harman[1,2]

[1]Department of Environmental Health and Engineering, Johns Hopkins University
[2]Department of Earth and Planetary Science, Johns Hopkins University

**Correspondence:** Ciaran Joseph Harman (charman1@jhu.edu)

**Abstract.** End-Member Mixing Analysis (EMMA) is a method of interpreting stream water chemistry variations and is widely used for chemical hydrograph separation. It is based on the assumption that stream water is a conservative mixture of varying contributions from well-characterized source solutions (end-members). These end-members are typically identified by collecting samples of potential end-member source waters from within the watershed and comparing these to the observations. Here we introduce a complementary data-driven method (Convex-Hull End-Member Mixing Analysis – CHEMMA) to infer the end-member compositions and their associated uncertainties from the stream water observations alone. The method involves two steps. The first uses Convex-Hull Non-negative Matrix Factorization (CH-NMF) to infer possible end-member compositions by searching for a simplex that optimally encloses the stream water observations. The second step uses Constrained K-means Clustering (COP-KMEANS) to classify the results from repeated applications of CH-NMF and analyzes the uncertainty associated with the algorithm. In an example application utilizing the 1986 to 1988 Panola Mountain Research Watershed dataset, CHEMMA is able to robustly reproduce the three field-measured end-members found in previous research using only the stream water chemical observations. CHEMMA also suggests a fourth and a fifth end-member can be (less robustly) identified. We examine uncertainties in end-member identification arising from non-uniqueness, which is related to the data structure, of the CH-NMF solutions, and from the number of samples using both real and synthetic data. The results suggest that the mixing space can be identified robustly when the dataset includes samples that contain extremely small contributions of one end-member – samples containing extremely large contributions from one end-member are not necessary, but do reduce uncertainty about the end-member composition.

## 1   Introduction

End-Member Mixing Analysis (EMMA) has been used to interpret observed stream water chemical concentration variability in terms of time-varying contributions from unknown end-member "sources", each supplying water with a constant concentration profile. This method has been applied in many different hydro-climatic and geology settings (e.g., Bernal et al., 2006; Hooper et al., 1990; Li et al., 2019; Liu et al., 2008a, 2017; Lv et al., 2018; Jung et al., 2009; Neill et al., 2011). EMMA has also been used to distinguish sources of dissolved organic matter in natural streams (Hur et al., 2006; Yang and Hur, 2014), specific conductance (Kronholm and Capel, 2015), and other combinations of stream water attributes that can be assumed to

mix conservatively (Barthold et al., 2011).

EMMA assumes that the chemical solute composition of stream water can be explained by the conservative mixing of a finite set of end-members (Hooper et al., 1990). These end-members, therefore, are the most extreme points of a simplex within which all stream water samples must lie (if the assumptions of the method are valid). End-members are identified by

collecting samples of candidate source-water from within the watershed; i.e., in addition to the "mixture" samples collected in the stream. The EMMA method assumes that 1) solutes used in the mixing model are conservative, 2) stream water consists of an identifiable number of end-member sources, 3) end-member compositions are distinct for at least one tracer, and 4) end-member compositions are spatiotemporally constant (or their variations are known or can be reduced by adding additional end-members) (Hooper et al., 1990).


Christophersen and Hooper (1992) suggested that "[u]nambiguous identification of the source solution compositions from the mixture alone is impossible". In a strict sense, this is likely true in that the underlying assumption (streamflow as a conservative mixture of invariant sources) is unlikely to be adhered to in a real watershed. However, recent advances in statistical learning methods suggest that some utility may exist in attempting to identify (perhaps not free of ambiguity) potential

source solution composition from the observed mixture alone (without additional candidate source-water samples) (Ding et al., 2008; Hyvärinen and Oja, 2000; Thurau et al., 2011). Here we propose a method, Convex-Hull End-Member Mixing Analysis (CHEMMA), which can in fact identify source solution compositions from the mixture alone. We will also present an analysis of the "ambiguity", or uncertainty, in the identified end-members.

It is worth distinguishing CHEMMA from previous applications of statistical learning methods (such as maximum likelihood estimation, Bayesian inference, and Markov Chain Monte Carlo, MCMC) to estimate uncertainties of end-member mixing analysis. Genereux (1998) presented a linear estimator for uncertainties in end-member concentration and mixing ratios. Carrera et al. (2004) achieved a similar approach by using the maximum likelihood method. By combining likelihood methods, Bayesian inferences, or probabilistic linear models with MCMC algorithm, Barbeta and Peñuelas (2017), Beria et al. (2020),

Delsman et al. (2013), and Popp et al. (2019) were able to acquire time-evolving uncertainty estimation. These contributions focus on quantifying uncertainty resulting from the use of field-sampled candidate end-members. In contrast, CHEMMA aims to infer the end-members themselves.

Stream water concentrations of different conservative solutes are naturally correlated. EMMA uses Principal Component

Analysis (PCA) to convert the naturally correlated stream water concentrations into a set of linearly uncorrelated variables (Christophersen and Hooper, 1992). Each new variable, which is called Principal Component (PC), is a linear combination of the observed stream water attributes. For a set of $n$ variables, PCA first requires standardized observations ($\mathbf{X}_{obs}$) by subtracting the mean and dividing by the standard deviation. Then it calculates a projection matrix $\mathbf{P}_{obs}$ (rows of which are eigenvectors of the correlation matrix), which transforms from observation space to PC space, by decomposing correlation matrix of $\mathbf{X}_{obs}$.

The transformed columns of $\mathbf{Y}_{obs}$ (representing the $n$ observations in the PC space) are uncorrelated each of which accounts for a portion of total variance (Christophersen and Hooper, 1992):

$$\mathbf{Y}_{obs} = \mathbf{X}_{obs}\,\mathbf{P}_{obs}^{T}. \tag{1}$$

Standardized end-member candidates $\mathbf{X}_{em}$ can be projected into the PC space by the same projection matrix $\mathbf{P}_{obs}$ and then converted in the transformed space as $\mathbf{Y}_{em}$ (Christophersen and Hooper, 1992):

$$\mathbf{Y}_{em} = \mathbf{X}_{em}\,\mathbf{P}_{obs}^{T}. \tag{2}$$

To find the parsimonious subset of appropriate end-members, EMMA subsequently takes the information provided by PCA to determine the approximate dimensionality of the stream water mixture and to screen end-members (Hooper, 2003; Liu et al., 2008a). In the PC space, appropriate end-member candidates ($\mathbf{Y}_{em}$) are selected by choosing ones that tightly bound the transformed observations ($\mathbf{Y}_{obs}$) (Christophersen and Hooper, 1992; Hooper et al., 1990; Hooper, 2003). Christophersen

and Hooper (1992) mathematically proved that one end-member more than the number of PCs is required to describe the rank of the stream water observation. However, the number of retained PCs is usually determined using a heuristic, such as using the number of PCs that explain at least $\frac{1}{n}$ proportion of the total variance because of the need capturing the variance (Hooper, 2003). In addition, Hooper (2003) suggests to examine the residual distribution pattern as an auxiliary technique for determining the dimensionality revealed by the data.


Limitations to this approach exist which can result in spurious or incomplete source identification (Delsman et al., 2013; Hooper, 2003; Valder et al., 2012; Yang and Hur, 2014). Specifically, the composition of a source cannot be determined unless candidate end-member measurements are obtained that are representative of it. In addition, determining the number of significant PCs , or the number of end-members, is subjective to some degree even with the aid of DTMM. EMMA is not able to deal

with non-conservative mixing if a non-linear structure is not provided to replace the current simplex structure (Christophersen and Hooper, 1992); therefore, only tracers that are believed to be approximately conservative should be included because they entered the stream, thereby altering their concentrations primarily by dilution rather than other mechanisms. Finally, another limitation involves uncertainties introduced by spatial and temporal variability in end-member concentrations that may cause additional difficulties (Delsman et al., 2013).


Here we focus on the first of these issues. In spite of EMMA's wide application (Ali et al., 2010; Bernal et al., 2006; Burns et al., 2001; Delsman et al., 2013; Hooper and Christophersen, 1992; James and Roulet, 2006; Jung et al., 2009; Li et al., 2019; Lv et al., 2018; Neal et al., 1992; Neill et al., 2011; Valder et al., 2012), no method exist to characterize missing or unmeasured end-members purely based on stream water observations other than using baseflow to characterize groundwater

(Liu et al., 2008b). Popp et al. (2019) came close, introducing a residual end-member that represents collective behavior of all other unobserved end-members, though it still requires some a-priori knowledge of "observed" end-members to initiate a Bayesian mixing model. In contrast, CHEMMA aims to identify the entire suite of end-member compositions and their associated uncertainties.

The CHEMMA method depends on the idea inherited from EMMA that the end-members are located near the most extreme
points in the "mixing" space of streamwater samples. Note that this does not imply that the concentration of any particular
solute is extreme in an end-member or that the end-member composition is even distinct for all solutes. Rather, it only implies
that the linear combination of concentrations in PC-space is extremal at the end-member. This suggests that we may be able
to interrogate the observational data projected in the end-member space to locate such extremal end-members even if no
individual samples fully represent that end-member. The approach we propose, CHEMMA, is a data-driven method to exploit
this possibility and to characterize end-members chemical composition as well as the associated uncertainty. The capabilities
of this method are demonstrated by an application to the 1986 to 1988 Panola Mountain Research Watershed dataset published
in Hooper and Christophersen (1992). We will further explore the robustness of this method using synthetic datasets generated
with three end-members.

## 2 Methodology

Convex-Hull End-Member Mixing Analysis (CHEMMA) applies the matrix factorization method, Convex-Hull Non-negative
Matrix Factorization (CH-NMF), along with the classification method, Constrained K-means Clustering (COP-KMEANS),
to determine end-member compositions under EMMA assumptions. The CH-NMF method provides a numerical iterative
algorithm to search for end-member compositions that optimally enclose the stream water observations in the PC space. The
CH-NMF algorithm is run many times because each iteration of the search can result in highly non-unique optima. We apply
the COP-KMEANS method to classify the CH-NMF numerical outputs into clusters. The centroid of each representing our
best estimate of an end-member.

### 2.1 Adaption of CH-NMF to the EMMA problem

The concepts of "convex combination" and "convex hull" connect CH-NMF with the idea of end-member mixing. A convex
combination is equivalent to a weighted sum. It is a linear combination of vectors where the weight associated with each vector
varies between zero and one, and the weights sum to one. If we construct a simplex, which means a highly dimensional poly-
tope, with some distinct vectors at its vertices, this simplex is a convex hull that encloses points within the hull to be a convex
combination of the vertices. Similarly, if we conservatively mixed distinct end-members, the stream water chemical concen-
tration observations can be a weighted sum of end-members with their contributions. The ideas of "convex combination" and
"convex hull" are mathematically identical to end-member mixing.

The CH-NMF method describes a general methodology of finding the most extreme points (end-members) that form a
simplex with $k$ vertices around the $n$-dimensional observation data cloud by searching for a convex hull that encloses the
data (Thurau et al., 2011).CH-NMF requires the rank $k - 1$ of the data needs to be first determined independently. PCA can
help with this. The top $k - 1$ PCs are retained as with EMMA using standardized (zero mean and unit variance) observations.
The CHEMMA algorithm does not entirely avoid this subjective choice of the number of end-members retained, and so does

not resolve this criticism of EMMA. The Diagnostic Tool of Mixing Models (DTMM) can also be used in conjunction with EMMA to determine the rank of the data. Next, the standardized data are projected into the 2D subspace spanned by two of the PCs (i.e., $PC_i$ vs $PC_j$, where $i \neq j$, $i < j$). Qualified points forming a convex hull around the projected data are marked at each pairwise 2D subspace. Finally, we interpolate between convex-hull vertices in each subspace to find $k$ vertices that define a $(k-1)$-dimensional mixing simplex. This simplex forms a convex-hull such that all the data points can be optimally approximated as convex linear combinations of them. The algorithm is summarized as follows:

---

**Algorithm 1:** CH-NMF algorithm (Thurau et al., 2011) adapted to the end-member identification problem given $m$ stream water observations of $n$ solutes

---

**Result:** $i^{th}$ standardized end-member composition $\boldsymbol{x}_{emi}^{n \times 1}$, and its contribution $\boldsymbol{h}_i^{m \times 1}$, $i = 1, 2, ..., k$

1. For each solute time series, subtract the mean ($\mu_{1:n}$) and divide by the standard deviation ($\sigma_{1:n}$) to obtain standardized observation matrix $\mathbf{X}_{obs}^{m \times n}$

2. Compute $d$ eigenvectors (PCs) $\boldsymbol{e}_1, ..., \boldsymbol{e}_d$, where $d = rank(\mathbf{X}_{obs}\mathbf{X}_{obs}^T) \leq n$

3. Project $\mathbf{X}_{obs}$ onto each of the $\binom{d}{2}$ 2D-subspaces spanned by pairs of PCs (similar form as Eqns. 1 & 2)

4. Mark all convex hull vertices for each projection plane and store in matrix $\mathbf{S}^{n \times p}$, where $p$ is the sum of the number of points found to make a convex hull in each projection plane.

5. Define end-member matrix $\mathbf{X}_{em}^{n \times k} = [\boldsymbol{x}_{em1}, \boldsymbol{x}_{em2}, ..., \boldsymbol{x}_{emk}]$ and let $\mathbf{X}_{em} = \mathbf{SI}$, then minimize $\|\mathbf{S} - \mathbf{SI}^{p \times k}\mathbf{J}^{k \times p}\|_F^2$, s.t. $\sum_i \boldsymbol{i}_j = 1, i_{ij} \in [0,1]$, and $\sum_i \boldsymbol{j}_j = 1, j_{ij} \in [0,1]$. Matrix $\mathbf{I}$ limits the end-member ($\mathbf{X}_{em}$) to be within the convex-hull construct by the stored extreme points $\mathbf{S}$, and $\mathbf{J}$ constrains those extreme points within the convex-hull formed by end-members.

6. Minimize $\|\mathbf{X}_{obs} - \mathbf{H}^{m \times k}\mathbf{X}_{em}^T\|_F^2$, s.t. $\sum_j \boldsymbol{h}_i = 1, h_{ij} \in [0,1]$, where $\mathbf{H}$ represents the fractional contribution of each end-member.

---

Given $m$ standardized stream water samples each with $n$ measured attributes $\mathbf{X}_{obs}^{m \times n}$ and $k$ desired end-members (Step 1, Figure 1 a), CH-NMF decomposes the correlation matrix of the observations to obtain at most $d$ PCs ($d$ is the maximum number of linearly uncorrelated variables), which is the same linear orthogonal projection as the Principal Component Analysis (PCA) method (in Step 2, from Figure 1a to Figure 1b, notice the changing distribution of the blue points). Instead of immediately reducing the dimensionality by discarding some PCs (as with EMMA), CH-NMF examines the distribution of $\mathbf{X}_{obs}$ in all of the subspaces spanned by PC pairs (Step 3, Figure 1b, light blue points) and marks the most extreme points (Figure 1b, red crosses) that construct the convex hull (Figure 1b, red lines) to store in $\mathbf{S}$ (Step 4). Then, a subset of $\mathbf{S}$, $\mathbf{SI} = \mathbf{X}_{em}$, is found as a convex combination of $\mathbf{S}$ (Step 5, Figure 1c, square vertices of the simplex) that minimizes the Frobenius norm $\|\cdot\|_F^2$ (the entry-wise Euclidean norm of the matrix). Inasmuch as $\mathbf{SI}$ may be any possible points within the convex-hull constructed using $\mathbf{S}$, $\mathbf{J}$ is needed to force $\mathbf{SI}$ to be chosen close to the convex-hull boundary. In practice, $\mathbf{I}$ and $\mathbf{J}$ are estimated iteratively using an optimization procedure until they converge (Eqns. 1 and 2 from Thurau et al. (2011)). Finally, the contribution $\mathbf{H}$ is found

by locating the convex combination of end-members that reproduces the data with minimal error (again using the Frobenius norm) (Step 6).

Step 5 is the essential step of the CH-NMF theory, and it is a modification of Convex Nonnegative Matrix Factorization (C-NMF) by adding a convexity constraint on $\mathbf{J}$ that ensures each component contributes a fraction between zero and one, with the sum of all fractions being one (Ding et al., 2008; Thurau et al., 2011). In the original setting of C-NMF, the $\mathbf{I}$ and $\mathbf{J}$ are naturally sparse if the vertex search is in PC subspaces (Ding et al., 2008). Adding the convexity constraint on $\mathbf{J}$ makes $\mathbf{J}$ an interpolation between each columns of $\mathbf{SI}$ (i.e., each end-member composition $\boldsymbol{x}_{em}$); however, the sparse nature of $\mathbf{I}$ remains (Thurau et al., 2011).

We could interpret the objective function of Step 5 (minimize $\|\mathbf{S} - \mathbf{SI}^{p \times k} \mathbf{J}^{k \times p}\|_F^2$) in three steps. First , the sparsity of $\mathbf{I}$ results in the end-member composition $\mathbf{X}_{em}$ close to a subset of the extreme observations ($\mathbf{S}$) projected in the PC subspace. Second, $\mathbf{J}$ makes other extreme observations in $\mathbf{S}$ to be expressed as a convex combination (interpolation) of $\mathbf{X}_{em}$. Third, minimizing the Frobenius distance between $\mathbf{S}$ and $\mathbf{X}_{em}\mathbf{J}$ guarantees end-member compositions $\mathbf{X}_{em}$ will be convex hull vertices because all other extreme points can be written as convex combinations of vertices, but not vice versa. As a consequence, a well-supported set of convex hull vertices tightly bound the observations and are as unique as possible, which satisfies the original EMMA assumption of finite set of distinct end-members. The sparse nature of $\mathbf{I}$ helps prevent overfitting because noise will tend to be concentrated on superfluous vertices without degrading identification of the others. The noisy end-members can be identified in the classification step given in the next section.

The constraint requiring that the end-members be a convex combination of the extreme observations implies that CH-NMF may not accurately identify end-members that are not a large fraction of any observation in the dataset. As the synthetic example shown in Figure 1 illustrates, the simplex formed by joining the CH-NMF end-members lies inside the shell formed by connecting the extreme points (red crosses in Figure 1c). Consequently, it is easier to identify end-members when more points lie on or near the hull itself so that the shape of the hull is clearly defined. In addition, if no samples are anywhere close to being "pure" representatives of an end-member, the apparent end-member identified by CH-NMF may lie closer to the data centroid than the true end-member. Methods to relax the constraint on Step 5 and better identify end-members distant from the data in the mixing space will be investigated in future work.

## 2.2 Quantifying the intrinsic uncertainty using COP-KMEANS

Each run of CH-NMF may yield different end-member estimates. This is because the complex structure of the high-dimensional stream water data results in a rough objective function surface (Step 5). CH-NMF runs with different initial search locations may fall into different local minima.

Depending on the structure of the data cloud, each run's end-members may be nearly identical (if the end-member is well-constrained by the dataset) or may vary widely. Poor identification may result if the data cloud lacks the clear planar boundaries that the CH-NMF algorithm looks for. It may also occur if more end-members are sought than the data can support or if an end-member is variable in time. The time-varying end-member "blurs" the planar boundaries and vertices. Alternatively, the observations may not sample the true mixing space sufficiently to identify an end-member in the space as a convex-hull vertex, perhaps because it never represents more than a small fraction of variance.

Even in the absence of these issues, the variability and uncertainty of the stream concentration observations will contribute to uncertainty in end-member identification. The variation in the CH-NMF-identified end-members can be assessed by running the CH-NMF analysis a large number of times and then using a clustering algorithm to extract the centroid and spread of areas consistently identified as an end-member. We use the COP-KMEANS variant of the K-means clustering algorithm, which allows us to require that end-members predicted from the same CH-NMF run must not be placed in the same cluster (Wagstaff et al., 2001). This is achieved by assigning a "cannot-link" constraint between every pair of candidate end-members generated by the same CH-NMF run. Apart from the "cannot-link" constraints, COP-KMEANS works identically to normal k-means clustering (Wagstaff et al., 2001). For each cluster identified by COP-KMEANS, we can qualitatively examine the spatial distribution of the associated end-members and quantitatively calculate the centroid and variance of the cluster.

## 2.3 Assessing the goodness of fit

There are several metrics that arise naturally from the CHEMMA framework that could be used to assess the goodness of fit of the inferred mixing subspace. The first and second are the centroid and within-cluster variance of each inferred end-member, which will tend to increase as the number of end-members increases. The third is the orthogonal projection distance from the observation space to the mixing subspace, which will be smaller when the end-member lies closer to the linear subspace where the rest of the data live. In this paper, we consider a new cluster to be tenable as a proper end-member if: 1) the spread of previously identified clusters remains similar or decreases, 2) the cluster itself has a reasonable variance, and 3) the orthogonal projection distances of previously identified end-members do not significantly increase after adding a new end-member.

We can also assess the degree to which CHEMMA and field-sampled end-members are "similar" to the stream chemical signatures. Field end-member candidate samples typically rely on a few grab samples (for example in Hooper et al. (1990), the groundwater was based on samples from a single well), which may insufficiently sample the overall source variability. CHEMMA end-members may provide a better idea of the time-space averaged chemical signature of a source than the field samples. One way to examine this is to look at the difference between an end-member's composition and its composition when projected into the reduced-rank $k - 1$ principal component subspace. This can be done for both field-sampled and CHEMMA end members. A summary measure of that difference is the Euclidean distance of the end-member from the reduced-rank

subspace. Where that distance is shorter, the end-member has a chemical profile that is aligned with that which is typically found in the stream. This distance can be calculated from the loadings on the remaining $n - k + 1$ principal components.

## 2.4 Example Python implementation

An example Python implementation of CHEMMA including the application to Panola Mountain data is presented in the next section. The code is available in a Jupyter Notebook on GitHub (https://github.com/Estherrrrrxu/CHEMMA). Updates can also be found from the GitHub page. The CH-NMF section uses a Python package, pymf.chnmf, detailed in Thurau et al. (2011). The COP-KMEANS section uses a Python package, COP-Kmeans presented in Babaki (2017).

## 3 Application to the Panola Research Watershed dataset

We applied CHEMMA to a test dataset of 905 samples of six solutes (alkalinity, sulfate, sodium, magnesium, calcium, and dissolved silica) collected from the stream in the Panola Mountain research catchment, Georgia, U.S. and described in Hooper et al. (1990). The six solutes were specifically selected to meet EMMA's assumption that their concentrations vary significantly across the watershed (Hooper et al., 1990). Hooper et al. (1990) suggested that the stream chemistry could be interpreted as a mixture of hillslope, groundwater, and organic soil horizon (organic) end-members, which are identified by sampling within the watershed. Hooper (2003) suggested that the rank of the data (Lower Gauge in Hooper (2003) dataset) is at least three. There was considerable evolution over time in the interpretation of these end-members (Hooper, 2001), but we will use the terminology from Hooper et al. (1990) to avoid confusion. Here we ask 1) does CHEMMA recover the same three end-members as Hooper et al. (1990) identified in field-sampling? and 2) does the data support the existence of additional end-members?

## 3.1 Results

We ran CHEMMA for three, four, and five end-member cases ($k = 3, 4, 5$) because two and three PCs account for $94\%$ and $97\%$ of the total variance, respectively . In order to capture the intrinsic uncertainty associated with the identified clusters, we calculated the mean and standard deviation (st. dev) for each case based on 100 CH-NMF runs (Table 1). CHEMMA was able to recover the three field-measured end-members reported by Hooper et al. (1990) (Figure 2, three blue stars). The mean of the three CHEMMA identified clusters (Figure 3 and Table 1) are very similar to the median concentration of the field-measured end-members (Table 2). The median concentration of the hillslope field sample (Table 2) has much lower alkalinity concentration compared with the mean concentration of the CHEMMA identified Green cluster (Figure 3 and Table 1); however, it is still within the cluster spread provided in Table 1.

The three CHEMMA end-members are also located closer to the subspace spanned by the $k - 1$ PC than the original three field-sampled end-members. The orthogonal projection distances are given in Table 3, and show that the CHEMMA end-members are more similar to the stream chemistry than the field samples, particularly for the groundwater end-member (field

sample distance: 0.814, CHEMMA sample distance: 0.450). The differences in the chemical signatures of the groundwater end-members and their projections in the data subspace are shown in Figure 4 (with concentrations given in standardized units, left for field samples and right for CHEMMA predictions). The CHEMMA end member's Alkalinity, $SO_4$, and Ca values in particular are much closer to that of the data subspace than the field-sampled end-member, which is indicated by the shorter distance from the original 6-D chemical profile in dots (blue for field samples and red for CHEMMA predictions) to the 2-D mixing space profile in flat caps (orange for field samples and green for CHEMMA predictions). Only for Si is the field-sampled value closer. After PCA dimension reduction, both field-sampled profile and CHEMMA-predicted profile are close in the standardized solute space. It is worth noting that CHEMMA does not require dimensional reduction; PCA is only needed to determine the number of end-members.

A fourth end-member could be robustly identified (Figure 2, four magenta diamonds) that explained more of the data variability. Hooper (2003) also suggested the existence of a fourth end-member. This end-member appeared to be a mixture of hillslope and groundwater in some ways but had relatively high alkalinity and silica concentration compared to those end-members (Figure 2 brown and navy axes). The fourth end-member captures variations along the third PC axis (Figure 3 d), which are not apparent in the 2D view (Figure 3 b).

The spread of all end-member clusters (generated by 100 runs of CH-NMF) was small when four were sought, but a fifth could not be clearly identified. As the number of end-members was increased from three (Figure 3a) to four (Figure 3b), the new cluster (cyan Cluster 4) was dense, while the other three clusters (green, blue, and red) remained at similar locations to those clusters identified in the three end-member case. Adding the fourth end-member reduced the spread of the previously identified three clusters in the PC subspace (Figure 3a and b and Table 1), suggesting that they could now be identified with less uncertainty. However, the inclusion of the fifth end-members (Figure 3c) did not further tighten the previously identified clusters; indeed, the fifth cluster was poorly defined (black Cluster 5). Except for the cyan cluster that generally decreased within cluster variation, the standard deviations of other clusters increased for both three and four end-member cases (Table 1).

The results in Figure 2 imply that identification of end members from the mixture alone may not be as "impossible" as Hooper and Christophersen (1992) suggested. CHEMMA is able to reproduce the three end-members that were identified in Hooper et al. (1990) as well as a fourth end-member, which explains more variation in the data.

This is not to say that the estimates provided by CHEMMA are "unambiguous", or even a complete set of contributing sources. CHEMMA identifies sources that can be found through their control on the boundary of the sample space. For example, sources that never supply the plurality of water but also that are never absent (or nearly never) may not be identified by CHEMMA, in that they never produce a "vertex"-like structure in the data cloud, nor do they constrain the location of a "face". Further work is needed to determine the limits on end-member identification for a given dataset.

## 3.2 Dimensionality and DTMM

For 4 CHEMMA end-member case in Table 3, the orthogonal projection distances of organic, hillslope, and groundwater end-members decrease/remain similar with 3 CHEMMA end-member case. Adding a fifth end-member significantly increases the projection distance of identified 4-th end-member. In addition, the dispersed cluster distributions in Figure 3c suggests that a fifth end-member may be spurious. We cannot rule out the possibility that it reflects only the noisy edges of the sample space, and so cannot be supported by the data. Indeed, CHEMMA does not come equipped with an objective criteria for determining how many end-members *can* be supported by the data. There are many mathematical methods, such as factor analysis and diffusion map spectral gaps, that could be used in parallel with CHEMMA to estimate data dimensions (Ashley and Lloyd, 1978; Coifman et al., 2008). It may be possible to use k-fold cross validation of CHEMMA itself to try to determine the best number of end-members. CHEMMA can also be used in conjunction with the approach already developed for EMMA to assess dimensionality: DTMM presented in Hooper (2003)). DTMM (Hooper, 2003) suggests choosing the smallest possible number of end-members that gives uncorrelated residuals resembling random noise. Any correlation structure in the residuals suggests a lack of fit in the model, which could be caused by (among other things) outliers and nonconservative solutes. An additional dimensionality (additional eigenvector to be retained) can be added until the residual structure is unseen or is not improved.

## 3.3 Uncertainty analysis

Because CHEMMA extracts end-members from the observations, the accuracy of the end-member's composition is influenced by a range of sources of variability and uncertainty, including how much noise exists from sample analysis error, how well the collected samples represent the full range of sources in the catchment, how many end-members we assume that (as discussed above), how unique the CH-NMF and COP-KMEANS analyses are, and how valid the assumptions are that end members are conservatively-mixed and time-invariant. For example, rare contributions from an end-member may result in the the dispersion of Cluster 3 in Figure 3b. Temporal variations of the end-member composition could produce the kind of variations seen in PC 3 in Figure 3d (Inamdar et al., 2013). Fortunately, CHEMMA itself may be a basis for exploring the effects of time-variability. For example, by partitioning the dataset into time periods (or hydrologic state, etc), the apparent temporal variability of end-members could be explored.

Sampling uncertainty is a more tractable issue for the present analysis. We can estimate the magnitude of this error using bootstrapping (resampling with replacement) (Efron and Tibshirani, 1994). We generated 1000 bootstrapped sets of the original Panola data, and ran CHEMMA on each of them. The end-members identified in these bootstrapped datasets showed relatively little scatter compared to the overall variance of the stream water concentrations (Figure 5), suggesting that they are robust with respect to sampling error. Even the organic end-member, which dominates a limited number of stream water samples (Figure 2, the few grey points towards the organic end-member) could still be identified with considerably small variance compared with the original solute variation (as shown in Figure 5). However, this poorly-represented end-member shows

many more outliers (end-member compositions substantially different from the best estimate) than the other two. Figure 5 also re-emphasizes that CHEMMA identifies end-members that exhibit collectively unusual combinations of concentrations (i.e., vertex-like structures in the overall data cloud). While many solute concentrations of CHEMMA predicted end-members are located towards extremal values of the observations, they need not be all individually extremes (e.g. the sulfate concentration of end-member 3, corresponding to the hillslope end-member, Figure 5 upper middle plot)."

To see how robustly the end-members could be identified with a smaller number of observations, we ran CHEMMA on bootstrapped subsets of the original data. These subsets represented from $5\%$ to $100\%$ of the original data size (905), and each subsetting experiment was repeated 1,000 times. Results are shown in Figure 6. For this particular dataset, the uncertainty is substantial when fewer than $40\%$ (362) of the original data are used, decreasing greatly from $40\%$ (362) to $60\%$ (543). Further improvements in robust identification with more samples are mainly in the less well-constrained organic end-member (Figure 6).

In addition, the overall number of samples may matter less than the number of samples that are either dominated by one end-member, or in which an end-member is entirely absent. Four of the varying effects of sampling uncertainty on CHEMMA are illustrated in Figure 6: 1) Some end-member constitutes, such as $SO_4$ in the groundwater end-member (End-member 2), and Alkalinity, Na, and Si in the hillslope end-member (End-member 3), are well identified regardless of whether $5\%$ (45) or $100\%$ (905) of the total available sample size is used; 2) For the well-represented groundwater and hillslope end-members, the uncertainty bounds do not vary as dramatically with sample size as they do for the organic end-member, which is less frequently important; 3) Even using the full dataset, some of the end-member constituents are not very well-constrained (e.g., $SO_4$ of the organic end-member/End-member 1 has a larger variance than the well-constrained end-members with sample size as small as 45; 4) Clusters of outliers (or multi-modality in the bootstrapped replicates) may suggest poorly-constrained end-members. For example, $SO_4$, Mg, and Ca in hillslope end-member/End-member 3 identified with sample sizes 45 and 90 exhibit clusters of outliers in their tails. These clusters are within the range identified with end-member 1 using larger sample sizes.

## 3.4 A synthetic exploration on model robustness

We also examined uncertainties arising from potential non-uniqueness of the CH-NMF and COP-KMEANS analyses. Intuitively, we can expect these to be greatest when the dataset lacks the vertex-like structures that the algorithm seeks to identify. In Figure 7, the "algorithm" standard deviation denotes the variability amongst 100 CH-NMF runs (in one CHEMMA run), and the "data" standard deviation represents the variability amongst 100 bootstrapped CHEMMA runs. The variability induced by instability of these algorithms is small compared to the overall variability of the dataset, but is much greater than that introduced by the sampling alone.

To explore this source of uncertainty further, we created a relatively simple synthetic dataset of "observations" of two Gaussian-distributed independent variables (X and Y) that can be represented as conservative mixtures of three "true" end-members. As Figure 8 shows, X and Y are chosen to center on the conservative mixing triangle's incenter. The variance of the

340 Gaussian distributions used to generate these data increases from case 1 to 6 in Figure 8. All marked "estimated" end-members are outputs from 100 CH-NMF runs, which represents the end-member variation during one CHEMMA run (Figure 8).

As expected, when the observations have a low variance compared to the spread of the end-members CHEMMA does a poor job at identifying the end-members. In the case with the tightest cluster, case 1, the estimated end-members are actually less variable than in the less tightly clustered case 2. This suggests that variations between applications of CH-NMF are sensitive 345 to the particularities of a dataset's extremal observations.

Between case 3 and case 4, the stability of the end-members identified by CH-NMF becomes much better, even though the distribution of observations in case 4 seems to have been barely constrained by the mixing space. There is sufficient structure for the algorithm to anchor three unique end-members (Figure 8 and Figure 9). However, the estimated end-members are biased toward the centroid of the dataset, and do not accurately characterize the end-members. As the observations fill more of the 350 conservative mixing space within the triangle (i.e. the convex hull), CHEMMA-identified end-members are closer to the true end-members.

Figure 9 confirms and expands the observations from Figure 7 and 8 that the major uncertainty of CHEMMA predicted end-members comes from sampling errors when the dataset has sufficient structure. For the synthetic dataset, the algorithmic uncertainty becomes insignificant when the data cloud just begins to be constrained by the end members. In case 4 in Figure 355 8), less than 1% of the random samples generated fell outside the mixing space (and were thus discarded). Note that it is the edges, not the vertices, that have affected the shape of the data cloud at this stage. This suggests that the CHEMMA algorithm does not require that there be "extreme" samples containing large contributions from only one end member (i.e., samples close to a vertex in the mixing space). Rather, it can detect mixing structure robustly when the dataset includes samples containing very small contributions of one end member, and intermediate contributions of another (i.e., samples close to an edge/face 360 of the mixing space, but far from a vertex). However, an end-member whose contribution is consistently low may not be effectively detected because it does not sufficiently affect the shape of the data cloud boundary to justify increasing the number of end-members sought (i.e., the number of principal components retained in the analysis *plus one*).

## 4 Conclusion

Here we have advanced a method of end-member mixing analysis that challenges Christophersen and Hooper (1992)'s assertion 365 that source solution compositions cannot be unambiguously determined from the mixture alone. The traditional EMMA method requires potential end-member source waters to be sampled in the field and compared to the data.

The method presented, Convex Hull End Member Mixing Analysis, or CHEMMA, uses a combination of recently-developed statistical learning techniques to infer streamflow end-members from the stream water solute concentration data structure. The end-members are estimated by fitting a simplex ($k$-dimensional polyhedron) to the data cloud and identifying the end members 370 with the vertices of the simplex. The method was tested by applying it to the Panola dataset of Hooper et al. (1990). CHEMMA was able to accurately reproduce the field-sampled end-members identified in the original study solely from the streamwater samples.

Two sources of uncertainty in the chemical profile of the identified end-members were evaluated. Algorithmic error (variations between applications of the CHEMMA algorithm) was estimated by re-running the algorithm multiple times on the same dataset. Sample error was estimated by bootstrapping the original dataset and re-running the CHEMMA analysis 1,000 times. The results demonstrated that the end-members in the Panola dataset were identified with relatively little variance compared to the overall variance of the data. More of the error was due to algorithmic error rather than sampling error.

Subsampling of the Panola dataset demonstrated the sensitivity of the CHEMMA method to the number of samples. The results suggested that estimates of the end-members may be uncertain when too few samples are available. When an end-member is the major component of only a small proportion of the sample set (as is the case with the organic end-member in the Panola dataset). Some end-member constituents were reliably identified with as few as 45 samples (e.g., $SO_4$ in the groundwater end-member, and Alkalinity, Na, and Si in the hillslope end-member), while others needed more than 500 samples to be identified with similar robustness (e.g., all the constituents of the organic end-member).

A synthetic dataset was used to examine how uncertainty in the end-member identification was related to the data structure. This showed that algorithmic uncertainty could be large when the fringes of the data cloud were far from the "edges" and constrained by the need to be a mixture of the end-members. That is, when all the samples contained a non-trivial portion of all the end-members, and no end-member dominated any one sample, the shape of the data cloud did not provide usable information about the end-members. This uncertainty dropped dramatically once the boundaries of the data cloud contacted the boundaries of the mixing space, and so at least a few samples contained nearly zero contribution from at least one end member. Notably, it was not necessary for some minimum number of samples to contain majority contributions from each end-member. However, estimates of the end-member composition were biased toward the data cloud centroid unless such extremal samples (i.e., ones that were almost entirely composed of one end-member) were present in the dataset.

CHEMMA makes it possible to investigate stream chemical dynamics in terms of end-members even when the samples of candidate source waters are not available. However, even where such samples are available (or could be collected in the future) CHEMMA may be a useful tool to augment the traditional approach in the following ways: 1) reducing subjectivity when selecting from field-measured end-member candidates by comparing them to CHEMMA-identified end-members; 2) serving as a check on missing sources by characterizing end-members that are not represented in field samples; and 3) helping target candidate end-member field sampling by suggesting source characteristics. However, the usefulness of CHEMMA is limited by the structure of the data in mixing space. As Figure 9 suggests, CHEMMA will fail for datasets in which all end members are present in all samples to some non-trivial degree. Samples in which an end-member is absent provide critical information, and strongly control the location of the face of the convex hull used to identify the other end members.

It should be noted that CHEMMA itself does not establish a systematic way to determine the appropriate number of end-members $k$ for which to search. This choice must be made independently. However, it is compatible with the DTMM method presented by Hooper (2003) that has been used to make this judgement in the past. DTMM (Hooper, 2003) was used to conclude that 1) the dimensionality of the Panola dataset is at least 3 (i.e., at least 4 end-members are required) and 2) the possible fourth source (end-member) may be weathering products containing calcium and magnesium. CHEMMA was able to identify

a fourth end-member with such a characteristic without running through DTMM analysis.

This method can be improved in a wide range of ways. Future work should focus on 1) applying quantitative methods to eliminate the subjective choice of $k$, such as the Akaike Information Criterion (AIC), or Bayesian Information Criterion (BIC, or Schwarz criterion) (see Kuha (2004)); 2) relaxing the constraints on the CH-NMF algorithm (e.g., forcing Algorithm 1, Step 5 to construct a "perfect" convex hull) so that extreme points in $\mathbf{S}$ also lie inside the simplex, allowing the method to better characterize end-members that are never a large fraction of any samples; 3) further exploring the data requirements and uncertainty of the method, including better understanding of the relationship between the stability of COP-KMEANS clusters, the temporal variability of end-members, and the number of samples; and 4) pre-conditioning a Bayesian CHEMMA with priors based on field end-member measurements.

*Code and data availability.* Both the example code and data are available in a Jupyter Notebook on GitHub https://github.com/Estherrrrrxu/CHEMMA (Xu Fei, 2020).

*Author contributions.* Xu Fei and Harman were responsible for conceptualization, methodology, and visualization. Xu Fei was responsible for investigation, formal analysis, and writing (original draft). Harman was responsible for funding acquisition, supervision, and writing (review & editing).

*Competing interests.* The authors have no competing interests to declare.

*Acknowledgements.* Panola stream solute chemistry data from Hooper and Christophersen (1992) was collected with the support of the USGS and is available at http://hiscentral.cuahsi.org/pub_network.aspx?n=385. Thanks to Rick Hooper for providing thoughtful feedback on the draft manuscript, and to Dr. Joost Delsman and one anonymous reviewer for their careful reading. This work was supported by NSF grant EAR-1654194.

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

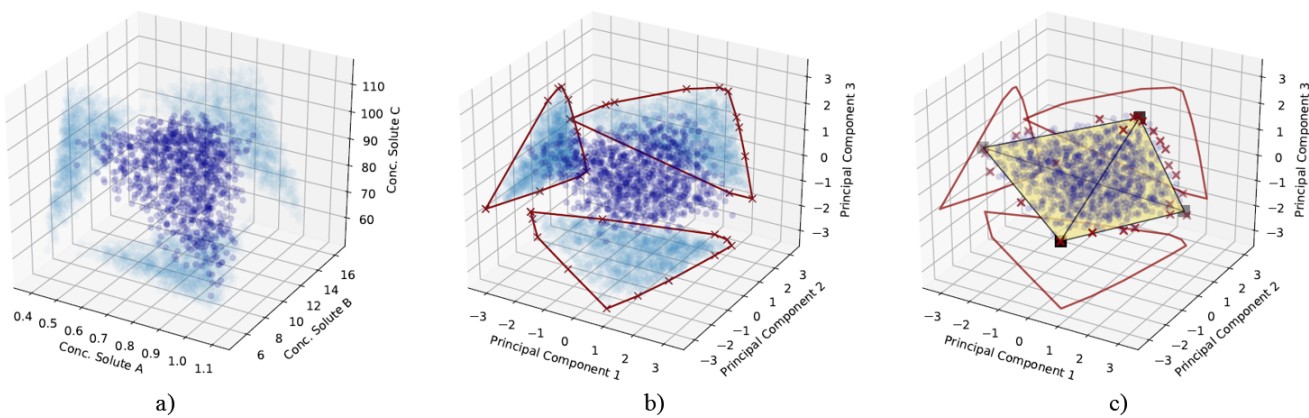

**Figure 1.** Illustration of the CH-NMF algorithm using a three dimensional space with four end-members. a) The standardized observations (dark blue) and its projection (light blue) on the observational space. b) The projected observations (dark blue) and its projection (light blue) on PC subspaces. The red crosses are the marked extreme points (**S**) that form a convex-hull (the red polygons) in each PC subspaces. c) Find the convex-hull (the black simplex) and its associated vertices (the $k$ vectors $\mathbf{x}_{emi}$) in the PC space, such that the vertices are convex combinations of the extreme points **S**, and the distance between the simplex and **S** is minimized. The red crosses are the same extreme points marked in b), but are projected back in the three dimensional PC space.

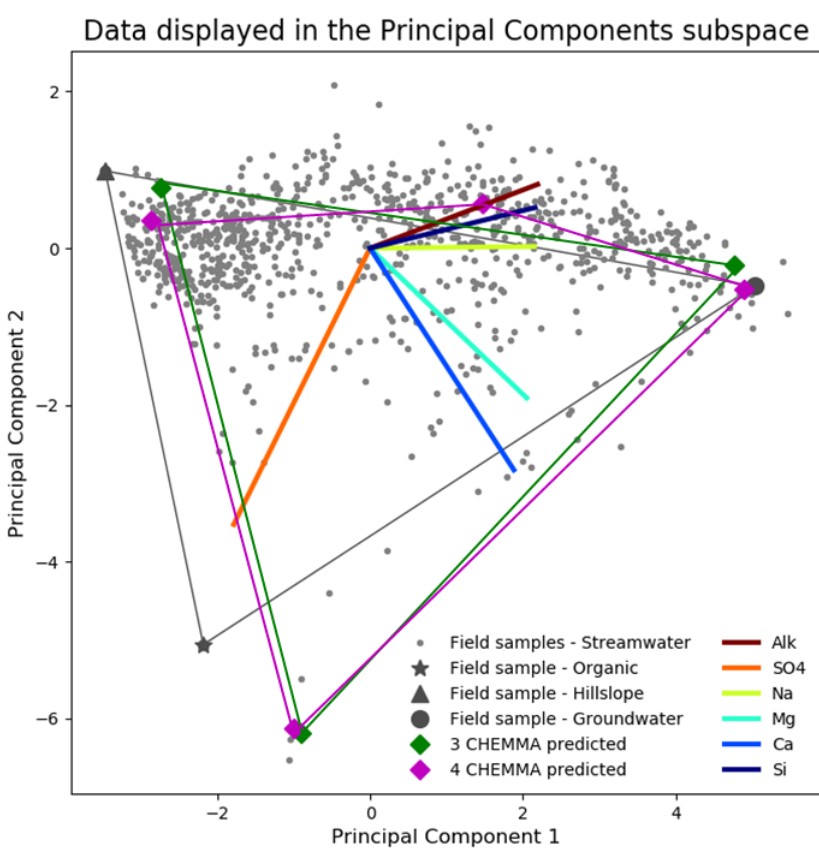

**Figure 2.** CHEMMA prediction (cluster centroids) for three end-member (blue diamonds) and four end-member (red diamonds) cases plotted in the PC2 vs. PC1 subspace. The colored lines that connect those predicted end-members indicate the convex hull projected into the two-dimensional PC subspace formed by those end-members. The observations (grey dots) inside of the convex-hull can be explained as linear combinations of the end-members. The colored lines in the center of the plot are the projected original solute axes in this PC subspace. Note that a three-dimensional subspace is required for four end-members.

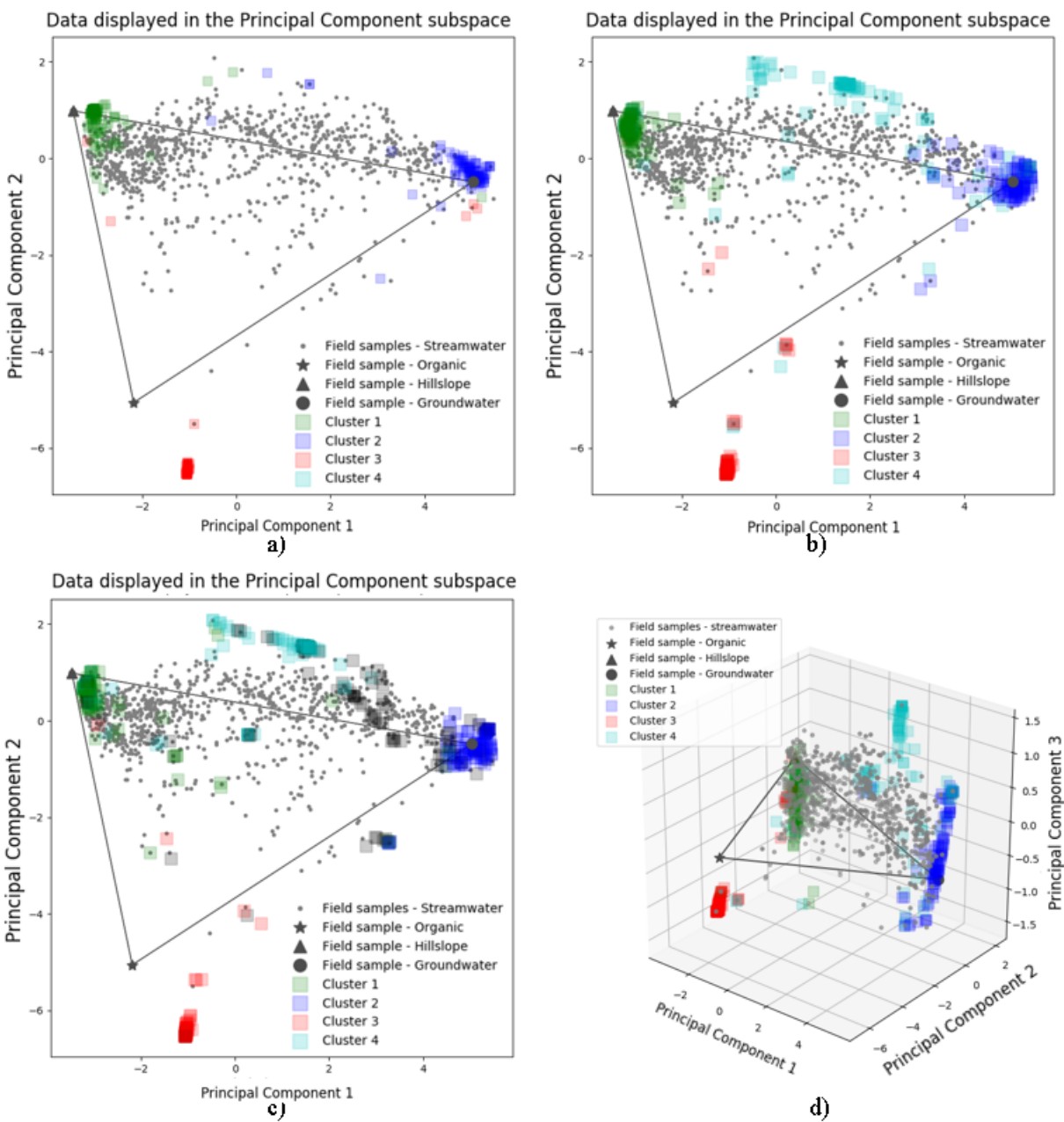

**Figure 3.** 100 random initialized CH-NMF runs result for three (a), four (b and d), and five (c) end-member cases. a - c are in the 2D PC2 vs. PC1 subspaces. d is in the 3D PC3 vs. PC2 vs. PC1 subspace. The color shade of each cluster reflects the concentration of the vertices at its location.

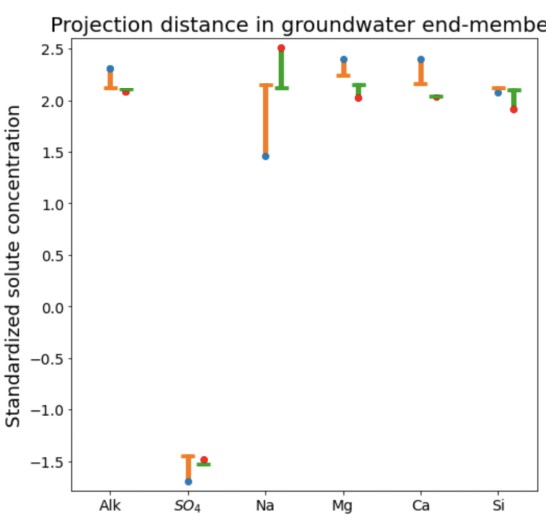

**Figure 4.** Comparison of orthogonal projection distance for field-measured and CHEMMA-predicted groundwater end-member solute concentrations (in standardized units). For each solute, the blue and red dots are the observed and CHEMMA-estimated end-member concentrations, respectively. The orange and green bars show how these concentrations change when the end-members are projected in the 2-D subspace (formed by retaining only the first two PC). The CHEMMA end-members lie closer to the 2-D subspace that PCA analysis suggests the data principally occupy, so their projection distances are smaller.

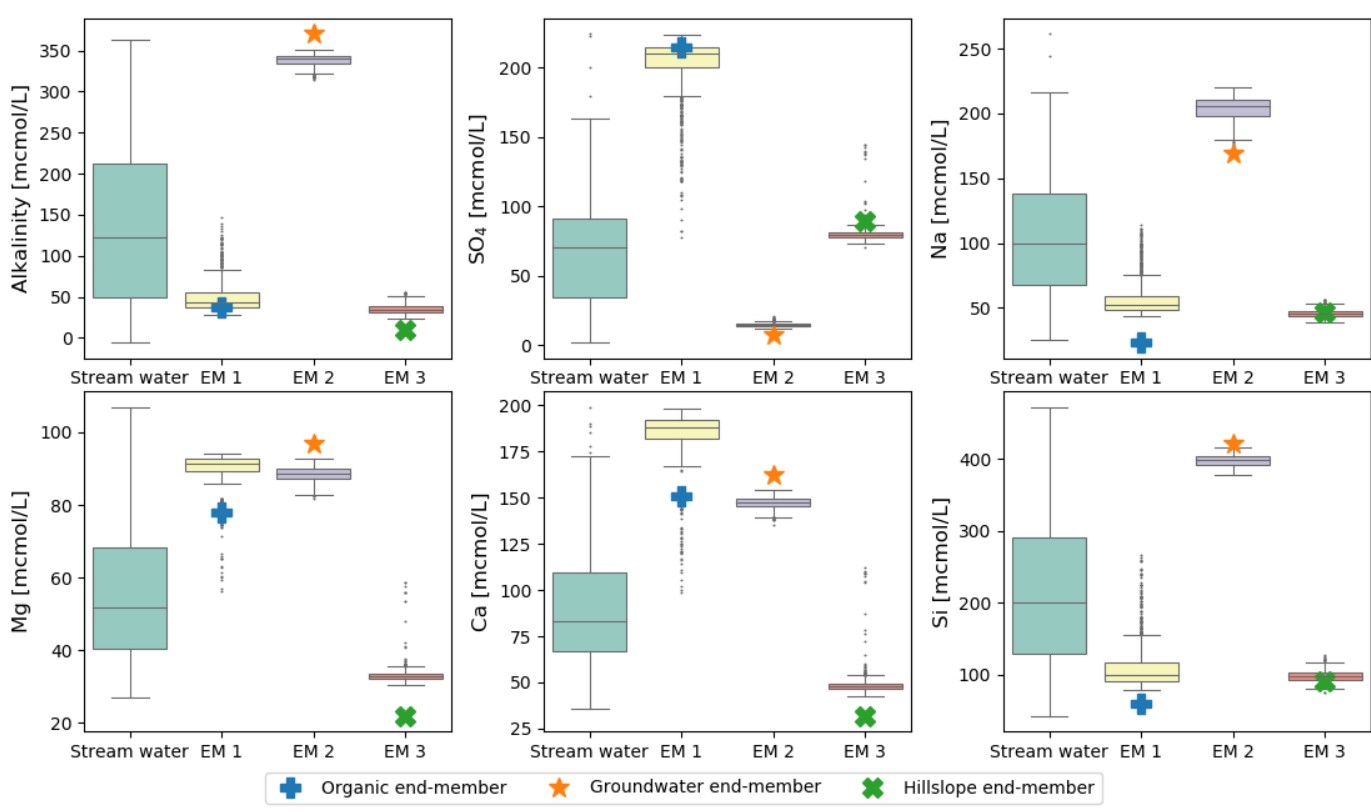

**Figure 5.** Uncertainties of CHEMMA predicted end-members (EM 1 to EM 3) compared with the total solute variances (Stream water). The four columns represent the stream water samples, and three CHEMMA identified end-members. Each CHEMMA end-member (EM 1 to 3, predicted without using field end-member samples) is matched with a field measured end-member based on the similarity of the concentration profile. The six subplots represents six stream water solute space.

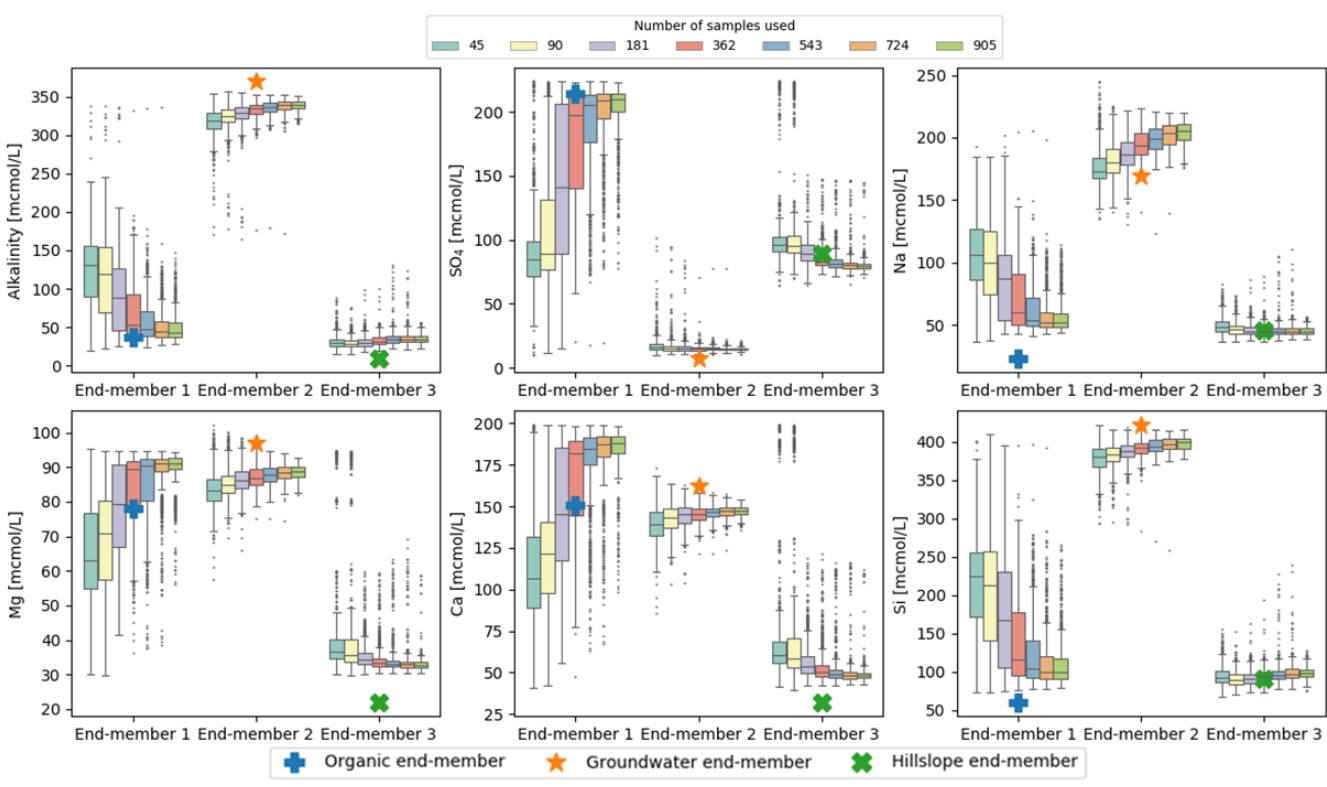

**Figure 6.** CHEMMA end-members predicted with varying sample size grouped by corresponding three field measured end-members. Each sample size box is drawn from 1000 bootstrap samples with the size of the number of sample used.

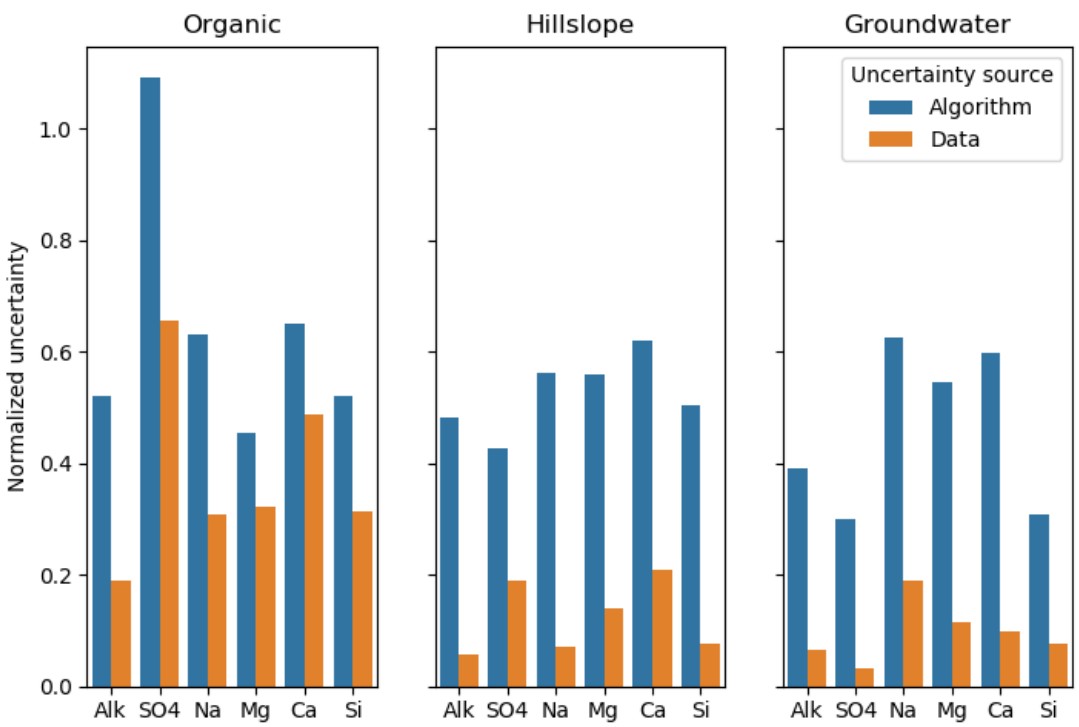

**Figure 7.** Normalized uncertainty of predicted end-members where the uncertainty sources are from algorithm and data groups using the Panola data. Algorithm and data are two groups that used the bootstrap method. Normalized uncertainties are estimated by dividing standard deviation of bootstrapped dataset over the standard deviation of streamwater solute measurements.

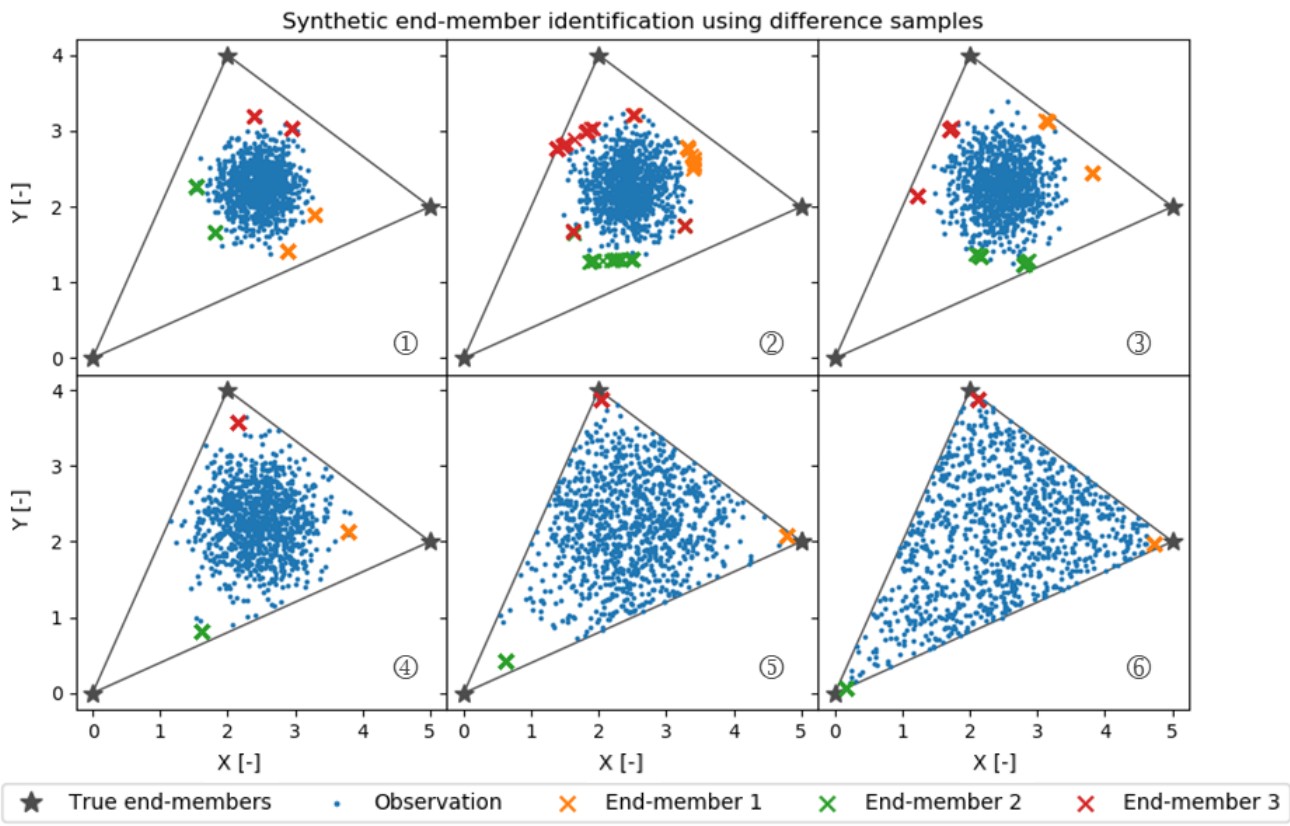

**Figure 8.** Synthetic random mixture (blue dots) generated by three fixed true end-members (grey stars). All cases (1 to 6) have the same number of samples (1000 samples) and are normally distributed around the inner center of the grey triangle. From case 1 to 6, the mixture occupies more of the convex mixing space as the standard deviation of the normal distribution used to generate those synthetic points increases.

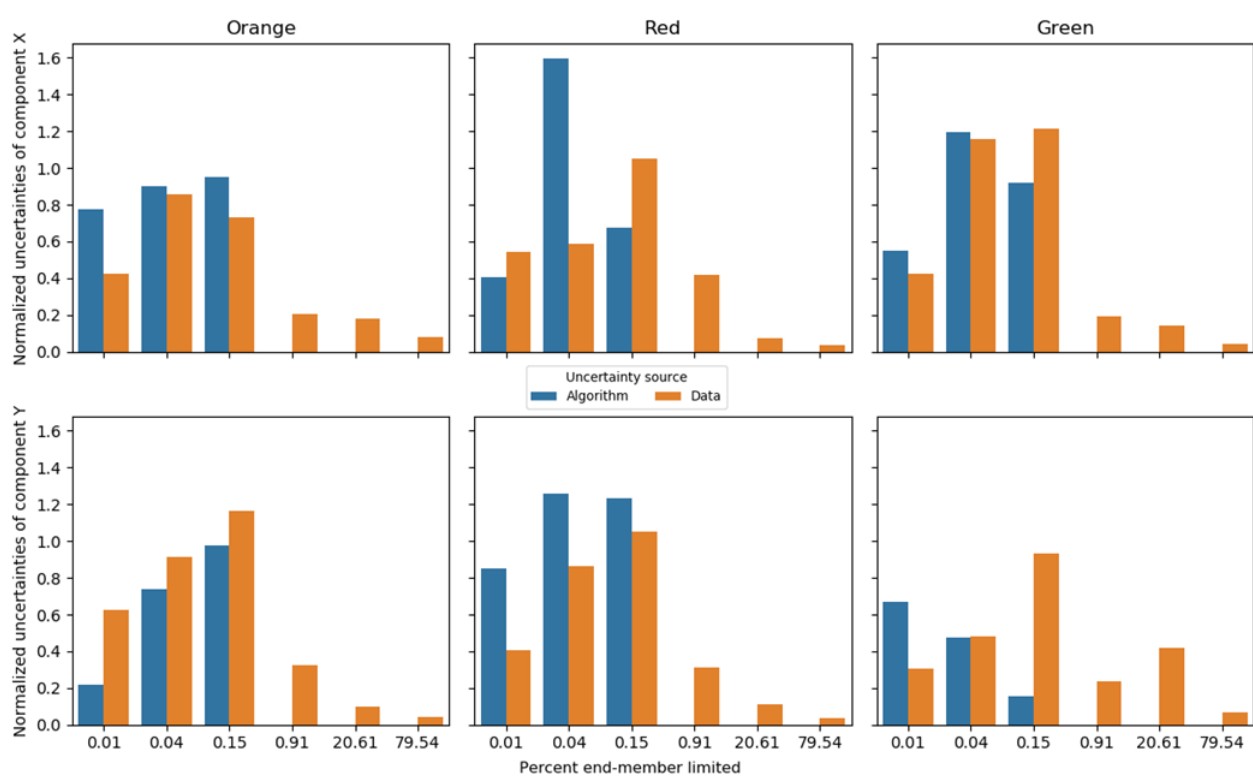

**Figure 9.** Standard deviation of predicted end-members with sources separated into algorithm and data groups using the synthetic data. Component X and Y represent the two synthetic measures, X [-] and Y [-], in Figure 8. X-axis percent end-member limited values are corresponding to synthetic case numbers in Figure 8. "Percent end-member limited" is a measure of the degree to which the synthetic data were constrained by the mixing space (it is the proportion of randomly-generated normally distributed samples that fell outside of the triangular constraint of the end-members, and were discarded). In each case, samples were continuously generated from the normal distribution with the given variance until 1,000 samples fell within the triangular constraint.

**Table 1.** The mean and standard deviation (st. dev) of each end-member cluster based on 100 random initialized CH-NMF runs. All values are in micromoles per liter. The cluster color indications correspond to Figure 3 a to c.

| # Clusters | | Alk | | SO4 | | Na | | Mg | | Ca | | Si | |
|---|---|---|---|---|---|---|---|---|---|---|---|---|---|
| | | Mean | St.dev | Mean | St.dev | Mean | St.dev | Mean | St.dev | Mean | St.dev | Mean | St.dev |
| Three | Red | 35.05 | 27.02 | 216.75 | 30.72 | 48.14 | 20.28 | 92.48 | 7.92 | 192.37 | 22.36 | 90.88 | 53.51 |
| | Blue | 348.04 | 12.16 | 14.11 | 2.82 | 214.87 | 21.88 | 90.35 | 4.64 | 151.26 | 9.93 | 405.86 | 23.55 |
| | Green | 33.43 | 32.27 | 77.45 | 12.60 | 44.70 | 20.01 | 32.03 | 5.84 | 47.14 | 10.75 | 100.34 | 55.85 |
| | | | | | | | | | | | | | |
| Four | Red | 32.86 | 12.33 | 219.71 | 17.57 | 46.66 | 9.91 | 93.50 | 2.44 | 193.92 | 15.11 | 87.25 | 28.64 |
| | Blue | 345.01 | 23.29 | 15.71 | 14.91 | 211.26 | 26.22 | 92.02 | 5.88 | 157.14 | 11.86 | 385.44 | 50.57 |
| | Green | 26.80 | 31.28 | 85.15 | 23.04 | 38.65 | 13.11 | 32.83 | 10.59 | 54.00 | 25.65 | 78.26 | 28.29 |
| | Cyan | 207.96 | 92.01 | 38.45 | 40.07 | 141.51 | 46.76 | 61.89 | 18.02 | 91.57 | 42.03 | 342.13 | 122.07 |
| | | | | | | | | | | | | | |
| Five | Red | 38.88 | 49.76 | 211.17 | 41.12 | 49.60 | 27.28 | 91.13 | 11.34 | 189.23 | 29.04 | 92.71 | 59.09 |
| | Blue | 344.76 | 21.77 | 15.88 | 14.39 | 211.90 | 30.95 | 92.44 | 5.63 | 158.67 | 12.07 | 390.34 | 40.03 |
| | Green | 29.62 | 33.35 | 85.37 | 13.38 | 42.52 | 17.68 | 33.40 | 6.83 | 52.32 | 16.99 | 84.20 | 29.38 |
| | Cyan | 171.83 | 77.99 | 40.85 | 33.32 | 123.60 | 44.11 | 54.77 | 15.08 | 75.69 | 29.17 | 329.06 | 138.29 |
| | Black | 253.45 | 107.65 | 44.10 | 47.45 | 161.55 | 58.00 | 75.81 | 17.47 | 125.51 | 38.38 | 278.05 | 123.41 |

**Table 2.** The median concentration of individual field measured end-members from Hooper and Christophersen (1992). All units are in micromoles per liter.

| Field individual samples | Alkalinity | $SO_4$ | Na | Mg | Ca | Si |
|:---:|:---:|:---:|:---:|:---:|:---:|:---:|
| **Organic** | 37 | 214 | 23 | 78 | 151 | 60 |
| **Groundwater** | 370 | 7 | 169 | 97 | 162 | 422 |
| **Hillslope** | 9 | 89 | 46 | 22 | 32 | 90 |

**Table 3.** End-member distance from observational plane to Principal Component subspace

|  | Organic | Hillslope | Groudwater | 4th | 5th |
|---|---|---|---|---|---|
| **Field Sample** | 1.217 | 0.298 | 0.814 |  |  |
| **3 EM CHEMMA** | 1.046 | 0.237 | 0.450 |  |  |
| **4 EM CHEMMA** | 0.816 | 0.223 | 0.482 | 0.377 |  |
| **5 EM CHEMMA** | 0.433 | 0.047 | 0.394 | 0.528 | 0.456 |