# Peer review of "A data-driven method for estimating the composition of end-members from stream water chemistry timeseries"

_Hydrology and Earth System Sciences, 2020_

## Short Comment (SC1) · 1 Jul 2020

End member mixing analysis is commonly applied to identify the dominant runoff producing sources of water. It employs tracers to determine the dimensionality of the hydrologic system. However, determination on the composition of the freshwater source is hard without observed candidate end-member measurements. This technical note provides a statistical method to effectively identify the end-member compositions and relative uncertainties. The scientific contribution is clear and the novelty of this method is enough.

[Figure]

But, I have a curiosity about the tracer set size and composition in this study, which may determine which and how many end members are identified. In my opinion, the number of end-member is highly sensitive to the tracer set size and composition. Why do you choose these tracers in your study?

———————————————————

---

## Referee Comment (RC1) · Joost Delsman (Referee) · 9 Jul 2020

Review of Xu Fei, E. and Harman, C.J. Technical Note: A data-driven method for estimating the composition of end-members from streamwater chemistry observations, HESSD.

Xu Fei and Harman present a method to extend classical end-member mixing analysis by deriving end-members purely based on stream concentrations. By repeatedly delineating the convex-hull around stream concentrations and then classifying results using a k-means clustering approach the method includes an uncertainty assessment of the resulting end-members. The method is successfully applied to the Panola Mountain Research Watershed data.

I consider the outlined method a valuable addition to the end-member mixing literature and recommend publication in HESS after revisions are made that are outlined below. I must admit I am not a math expert and have not checked the given formulae.

Current application of end-member mixing approaches involves the disentangling of stream water concentrations based on pre-conceived 'end-members', or origins of stream water. These end-members are sampled, and hopefully span the spread of stream water concentrations, thus enabling the calculation of flow route proportions. Xu Fei and Harman propose a method that calculates possible end-member concentrations from stream water alone. While this can indeed be very useful in practice, it also defies the purpose of an end-member mixing analysis to some extent: as a hydrologist, we are interested in where the water comes from that makes up the stream, not in its concentrations per se. I am sure the authors agree here, but I miss more discussion in the manuscript on the practical use of the proposed method in a hydrological analysis. Given the end-goal of finding water sources instead of end-member concentrations, where does their method come into play? As a first step, defining possible concentration profiles of end-members, after which you take to the field to 'find' these end-members? Or as a check of a more classical end-member mixing application: are end-members missed? Or vice-versa, as the proposed approach cannot handle end-members that are located outside the convex-hull of stream water concentrations.

Further, I would encourage the authors to provide more discussion of the uncertainty calculation of their method (how certain is the calculation of the convex hull), and how it relates to uncertainty in end-member mixing applications (how clearly defined is a single end-member, how time-variant is its concentration). On l197 you allude to some work you did on this, but this did not make it to the manuscript?

Overall, the manuscript would benefit from a thorough spelling and grammar check.

Minor comments:

l11: stream̲water

l30: concentration̲s: what you mean is the concentration of different solutes are correlated

l33: dividing b̲y the standard deviation. It doesn't require it per se, but yields better results

l34: transforms  from  observation space to  PC space

l35: and each of which accounts. What which? You are referring to the columns of Y? Why not just speak of the principal components?

l45: refer to diagnostic tools of (Hooper, 2003), as they propose a more formal analysis to assess the rank of the data. Hooper also finds evidence of a fourth end-member by the way… see around l173

l46: After _thus_ subjectively determin_ing_ the number of… Not sure what the purpose of this entire sentence is by the way, seems like it could be skipped altogether.

l50: also: spatial and temporal variability in end-member concentrations

l55: In spite of EMMA's wide application

l81: to  end-member mixing

l86: Only the top k PCs are retained? So while you criticize this subjectivity in lines 45-47, your method necessitates the same step? Please elaborate on this.

l87: Is there a reason why the analysis must be performed successively in 2D, and not in ND?

l89: "convex-hull": why here in quotes? This term has been used throughout the manuscript. Move its definition to its first use.

l91: algorithm: also define k and d. And: how are they different?

l123: This is quite a central drawback of the method. Can some directions / ideas already be given? Maybe a hybrid between chemma and field sampling for em signatures? Using the time-variance might also provide a way forward, searching for periods where certain end-members dominate and are thus better characterized from stream water concentrations.

l144: Elaborate on 'uncertainty' versus spread of local minima of convex hull. are these really the same thing?

l149: What is a 'reasonable' variance? Is there a suitable metric? Provide guidance.

l197: "Fortunately, CHEMMA itself provides a tool for exploring some of these sources of uncertainty. By partitioning the dataset into time periods (or hydrologic state, etc), the temporal variability of end-members could be explored" What is this statement based on?

On the python code:

Consider making your code available as an importable python module through pip and/or conda.

Why is Figure 2 different on github code? location 4[th] endmember? accessed 1/7/2020:

[Figure]

References

Hooper, R. P. (2003). Diagnostic tools for mixing models of stream water chemistry. *Water Resources Research*, *39*(3), 1–13. https://doi.org/10.1029/2002WR001528

---

## Referee Comment (RC2) · Anonymous Referee #2 · 13 Jul 2020

When I read the manuscript on the first time, I was thrilled, as this is something I have been waiting for many years to come out. If we could determine the chemical composition of end-members using streamflow chemistry alone, end-member mixing analysis would be significantly improved and revived. After I read it for a couple of times, I found the fundamental idea is still intriguing, but the assumptions the main method, namely CHEMMA, is based on may be flawed and cause significant uncertainties on the modeling results. A conceptual set-up of why and how this modeling would work could be strengthened. Readability could be improved as well, particularly in regard to some

mathematical details and their connection/implication with/in the hydrologic questions being investigated. Remember that most of readers who are interested in this study are hydrologists not mathematicians.

Major Comments:

The main approach is to use Convex-Hull Non-negative Matrix Factorization (CH-NMF) to infer possible end-member compositions by searching for a simplex that optimally encloses the stream water observations. The assumption for this is, based on authors, that end-members are located near the most extreme points that bound the observations in "mixing" space. From this assumption, it is clear that a simplex is basically determined by the data structure of observations, in other words, the shape of the sample cloud. What if one or more extreme points are missing in our observations? This could happen if samples are collected sparsely or only on certain hydrologic conditions/seasons that do not contain extreme samples (samples with extreme concentrations for at least one solute). The number of samples could also change how samples are distributed. With the same data set, can similar results (with reasonable uncertainties) be obtained from subsets of samples with varying number of samples that are randomly selected?

There is a lack of conceptual set-up where this study came from and where it goes in relation to existing tools in EMMA, particularly the diagnostic tools of mixing models (DTMM; Hooper, WRR, 2003). In one study, Christopherson and Hooper (WRR, 1992) specifically concluded that "Unambiguous identification of the source solution compositions from the mixture alone is impossible; thus, it is necessary that potential source solutions be derived from independent measurements." I do not mean this conclusion cannot be challenged, but the rationale must be stated clearly and explicitly, possibly using a conceptual set-up. Also, what is its relation with DTMM? Will the current study be supplemental or a substitute to DTMM in regard to the number of end-members? Can DTMM actually help to enhance CHEMMA and how?

The study used data collected in late 1980s. That is okay but what I am concerned is about the conservativity of all six solutes. How can we be convinced if all six solutes are conservative? If any of those is not conservative, the results of CHEMMA would be different. In my opinion, this is where DTMM may be able to help. Also, isn't it interesting to compare the number of end-members acquired using CHEMMA to DTMM?

Minor Comments:

L18: Before the first reference, add "e.g.,". Many classical references on EMMA were not actually cited.

L24: This statement should refer to conservative solutes.

L28: The second one is no longer a hypothesis or assumption because of the diagnostic tools of mixing models by Hooper (2003); See Liu et al. (WRR, 2008) for a demonstration and how this was addressed.

L30: "Streamwater concentration are naturally correlated." It is true if you refer to conservative solutes; otherwise it is an ill statement. Use two words "stream water" instead of one word "streamwater". Also, use plural for "concentration".

L31: Need at least one reference (e.g., Christopherson and Hooper, 1992).

L33-35: Multiple issues here. (1) Is Pobs actually eigenvectors? If so, use a parenthesis to annotate so; otherwise explain what it is and how to calculate it. (2) Get rid of the redundant "the". (3) My understanding is that once a standardized data set is used, a correlation matrix is decomposed rather than covariance matrix. Check if this is correct.

L36: If P are indeed eigenvectors, cite Christopherson and Hooper (1992) for the equation.

L41-42: Cite Hooper (WRR, 2003).

L45: True traditionally but not after DTMM is developed. See Liu et al. (WRR, 2008, 2017) as examples.

L51-52: Not true with DTMM.

L52-53: True but DTMM can help identify conservative solutes so that users can use only conservative ones. I mention this because I think your study is also based on mixing of conservative solutes. This should be stated/defined earlier in your text.

L60: Need to specify "extreme points". I think you refer to "extreme points of stream water samples".

L64: I think you mean "end-members' composition".

Result 2: Eigenvectors and PCs are different. PCs are calculated based on eigenvectors and observed concentrations.

Result 3: Is it specified anywhere how to project mathematically?

Result 4: Will the dimension of S differs from one projection plane to another?

Result 5: Is X expression actually [[xem1], [xem2], ..., [xemk]], as each xemi has a dimension of n by 1?

L93: I still think it is correlation matrix not covariance matrix. Also, what you mean here is eigenvectors not PCs.

L94: Spell out PCA as it appears for the first time.

L102: Specify the constraints, each between 0 and 1 with sum of all to be 1.

L125: I think "equifinality" is part of your talking here. Why not citing "equifinality" directly? It is a common term that hydrologists are very familiar with.

L186-206: Need to indicate where this modeling will lead to and how it may work together with DTMM.

---

## Author Comment (AC1) · 11 Aug 2020

Thank you for your encouraging comment. We acknowledge that dataset may largely influence the result of identified end-members. At the current stage, we used a well-studied set of tracers from Hooper et al. (1990) as an example to demonstrate the capability of CHEMMA.

References

Hooper, R. P., Christophersen, N., & Peters, N. E. (1990). Modelling streamwater

chemistry as a mixture of soilwater end-members—An application to the Panola Mountain catchment, Georgia, USA. Journal of Hydrology, 116(1-4), 321-343.

---

## Author Comment (AC2) · 2 Sep 2020

**Response to Referee #1**

Xu Fei and Harman present a method to extend classical end-member mixing analysis by deriving end-members purely based on stream concentrations. By repeatedly delineating the convex-hull around stream concentrations and then classifying results using a k-means clustering approach the method includes an uncertainty assessment of the resulting end-members. The method is successfully applied to the Panola Mountain Research Watershed data.

I consider the outlined method a valuable addition to the end-member mixing literature and recommend publication in HESS after revisions are made that are outlined below. I must admit I am not a math expert and have not checked the given formulae.

**Thank you for your concise summary and encouraging comment. We aim to provide a method that is both well-established in mathematical foundations and easy to understand for the general hydrology community. We have adopted some of your comments to improve the readability, particularly in the method section.**

Current application of end-member mixing approaches involves the disentangling of stream water concentrations based on pre-conceived 'end-members', or origins of stream water. These end- members are sampled, and hopefully span the spread of stream water concentrations, thus enabling the calculation of flow route proportions. Xu Fei and Harman propose a method that calculates possible end-member concentrations from stream water alone. While this can indeed be very useful in practice, it also defies the purpose of an end-member mixing analysis to some extent: as a hydrologist, we are interested in where the water comes from that makes up the stream, not in its concentrations per se. I am sure the authors agree here, but I miss more discussion in the manuscript on the practical use of the proposed method in a hydrological analysis. Given the end-goal of finding water sources instead of end-member concentrations, where does their method come into play? As a first step, defining possible concentration profiles of end-members, after which you take to the field to 'find' these end-members? Or as a check of a more classical end-member mixing application: are end-members missed? Or vice-versa, as the proposed approach cannot handle end-members that are located outside the convex-hull of stream water concentrations.

**We appreciate your suggestion in discussing practical applications of CHEMMA. In the revision, we also adopted the comment from Referee #2 and added a paragraph after l200:**

**"*For most hydrologists, end-member analysis is used to identify the water sources, and toward that purpose CHEMMA may be useful in the following ways: 1. CHEMMA may be used to reduce subjectivity when selecting from field-measured end-member candidates by comparing them to CHEMMA end-members; 2. CHEMMA may identify end-members that have not been sampled in the field, which may serve as a check for missing sources; 3. CHEMMA end-member compositions may help hydrologists ask better questions and provide guidance for field sampling by suggesting source characteristics; 4. CHEMMA can be used in conjunction with the Diagnostic Tool of Mixing Models (DTMM, developed by Hooper (2003)). DTMM is used to assess the tracer conservation, and mixture rank. CHEMMA can be enhanced by using DTMM analysis to select conserved tracers for analysis. The robustness of CHEMMA end-members also serve as a check for DTMM-determined rank of mixture.***"**

Further, I would encourage the authors to provide more discussion of the uncertainty calculation of their method (how certain is the calculation of the convex hull), and how it relates to uncertainty in end-member mixing applications (how clearly defined is a single end-member, how time-variant is its concentration). On l197 you allude to some work you did on

this, but this did not make it to the manuscript?

Overall, the manuscript would benefit from a thorough spelling and grammar check.

**Thank you for your suggestion. The uncertainty analysis within this technical note is limited to discuss the general intrinsic uncertainty introduced by the CH-NMF algorithm. However, much more work is required to dissect the total uncertainty arising from factors like the time-variability of the end-members and the algorithm's response to data uncertainty. We believe this is beyond the scope of a Technical Note introducing the approach. Therefore, we have decided to leave detailed analysis regarding uncertainty in practical applications for future work.**

Minor comments:

l35: and each of which accounts. What which? You are referring to the columns of Y? Why not just speak of the principal components?

**Thank you for pointing out this. "which" refers to $Y_{obs}$, which is the observation matrix projected onto the principal component space. The principal components are the rows of the projection matrix P.**

l45: refer to diagnostic tools of (Hooper, 2003), as they propose a more formal analysis to assess the rank of the data. Hooper also finds evidence of a fourth end-member by the way… see around l173

**Thank you for highlighting this. Hooper did not explicitly identify the composition of the fourth end-member (Hooper, 2003). We have edited the manuscript to reflect his speculation on its existence by adding the following sentence at l172:**

 *"Hooper (2003) also suggested the existence of a fourth end-member."*

l46: After thus determining the number of… Not sure what the purpose of this entire sentence is by the way, seems like it could be skipped altogether.

**Here we intended to differentiate the rank of mixture idea from EMMA with the mathematically defined rank *d* (refer to comment l91) from CHEMMA.**

l50: also: spatial and temporal variability in end-member concentrations

**Thank you for the suggestion. We have added a fourth point to address this comment:**

 *"4) uncertainties introduced by spatial and temporal variability in end-member concentrations cause extra difficulties."*

l86: Only the top k PCs are retained? So while you criticize this subjectivity in lines 45-47, your method necessitates the same step? Please elaborate on this.

**Thank you for identifying this confusion. We have added a sentence in the manuscript to acknowledge that the subjective selection of the number of end-members is not completely resolved by the proposed method:**

 *"The CHEMMA algorithm does not entirely avoid this subjective choice of the number of end-members retained, and so does not resolve this criticism of EMMA."*

l87: Is there a reason why the analysis must be performed successively in 2D, and not in ND?

**Thank you for noticing this technical detail. The search of finding the convex-hull is done in _all_ 2D PC projections, not necessarily successive and limited to k retained PCs. The 2D search is a way to subsampling the vertices set and it gives the most "greedy" search result because one needs at least 2D to draw lines/planes (or other linear structures in higher dimension) and to find the linear intersections (candidate end-members) that bound the observation. In addition, searching in 2D subspaces not only preserves detailed structure of the convex hull for further simplification, but also is efficient in terms of computation cost (Thural et al., 2011). We currently have not added a detailed explanation of this into the manuscript but the issue is addressed in the literature cited (de Berg et al, 2000) .**

l91: algorithm: also define k and d. And: how are they different?

**Thank you. d is the rank of the observation covariance matrix, as defined in Step 2 of Algorithm 1, whereas k is the number of eigenvectors to retain (which is stated in l86 as last comment shown). We have added "where" before the definition of d to make it stand out more. We have also changed sentence a) at l93 to _"CH-NMF decomposes the correlation matrix of the observations to obtain at most d PCs (d is the maximum number of linearly uncorrelated variables)"._**

l123: This is quite a central drawback of the method. Can some directions/ideas already be given? Maybe a hybrid between chemma and field sampling for em signatures? Using the time-variance might also provide a way forward, searching for periods where certain end-members dominate and are thus better characterized from stream water concentrations.

**Thank you for this insightful comment. It is certainly true that any method that relies exclusively on the stream water samples will face this challenge. We have shortly discussed this drawback both in the new added paragraph in l200 (in response to one of the above comments) and in the discussion part of the manuscript (l202 - l205). We have added a couple of sentences to address these suggestions in the discussion section.**

**After modified l199: "_The temporal end-member dominance may further deepen our understanding of stream water characterization._"**

**After l206: "_4) using individual field end-member measurements to inform CHEMMA._"**

l144: Elaborate on 'uncertainty' versus spread of local minima of convex hull. are these really the same thing?

**Thank you for your suggestion. Total uncertainty includes uncertainty due to the spread of local minima of convex hull as well as uncertainty due to sampling. Here we only examine the uncertainty arises from the algorithm. The complete examination of uncertainty is left for further work.**

l149: What is a 'reasonable' variance? Is there a suitable metric? Provide guidance.

**Thank you for this valuable comment. We have not identified a suitable metric at the stage. The current "reasonable" variance remains a subjective choice. We acknowledge the subjective here and we have modified the sentence in l202 from "_optimize the model complexity_" to "_eliminate the subjective choice of k_" for clarity.**

l197: "Fortunately, CHEMMA itself provides a tool for exploring some of these sources of uncertainty. By partitioning the dataset into time periods (or hydrologic state, etc), the temporal variability of end-members could be explored" What is this statement based on?

**Thank you. This statement is intended to provide suggestion for ways to explore temporal variability. We have modified the sentence to clarify the intent:**

**"*CHEMMA itself may provide a tool for exploring some of these sources of uncertainty. For example, by partitioning the dataset into time periods (or hydrologic state, etc), the temporal variability of end-members could be explored.*"**

On the python code:

Consider making your code available as an importable python module through pip and/or conda. Why is Figure 2 different on github code? location 4th endmember? accessed 1/7/2020:

[Figure]

References

Hooper, R. P. (2003). Diagnostic tools for mixing models of stream water chemistry. *Water*

*Resources Research*, *39*(3), 1–13. https://doi.org/10.1029/2002WR001528

**Thank you for your suggestion. We will make a python module in the near future.**

**In the technical note, Figure 2 is made by using the centroids of the clusters from Figure 3 a and b. The Python code only shows one run of CH-NMF and what you saw is the algorithm find another 4th end-member in the space. By repeatedly run the CH-NMF and using the COP-Kmeans to classify the clusters, this occasional capture of "wrong" convex-hull vertex can be eliminated.**

**We will add a section in the Jupyter notebook to reproduce Figure 2.**

**Thank you for your careful reading. We have adopted the following grammatic comments.**

l11: streamwater

l30: concentrations: what you mean is the concentration of different solutes are correlated

l33: dividing by the standard deviation. It doesn't require it per se, but yields better results

l34: transforms  from  observation space to  PC space

l55: In spite of EMMA's wide application

l81: to  end-member mixing

l89: "convex-hull": why here in quotes? This term has been used throughout the manuscript. Move its definition to its first use.

**References**

**Christophersen, N., & Hooper, R. P. (1992). Multivariate analysis of stream water chemical data: The use of principal components analysis for the end-member mixing problem. Water Resources Research, 28(1), 99-107.**

**de Berg M, van Kreveld M, Overmars M, Schwarzkopf O (2000) Computational geometry. Springer, Berlin, Heidelberg**

**Hooper, R. P. (2003). Diagnostic tools for mixing models of stream water chemistry. Water Resources Research, 39(3).**

**Hooper, R. P., Christophersen, N., & Peters, N. E. (1990). Modelling streamwater chemistry as a mixture of soilwater end-members—An application to the Panola Mountain catchment, Georgia, USA. Journal of Hydrology, 116(1-4), 321-343.**

**Thurau, C., Kersting, K., Wahabzada, M., Bauckhage, C., 2011. Convex non-negative matrix factorization for massive datasets. Knowl. Inf. Syst. 29, 457–478. https://doi.org/10.1007/s10115-010-0352-6**

---

## Author Comment (AC3) · 2 Sep 2020

**Referee #2**

When I read the manuscript on the first time, I was thrilled, as this is something I have been waiting for many years to come out. If we could determine the chemical composition of end-members using streamflow chemistry alone, end-member mixing analysis would be significantly improved and revived. After I read it for a couple of times, I found the fundamental idea is still intriguing, but the assumptions the main method, namely CHEMMA, is based on may be flawed and cause significant uncertainties on the modeling results. A conceptual set-up of why and how this modeling would work could be strengthened. Readability could be improved as well, particularly in regard to some mathematical details and their connection/implication with/in the hydrologic questions being investigated. Remember that most of readers who are interested in this study are hydrologists not mathematicians.

> Thank you for recognizing the value of our work and providing suggestions on improving the quality of this manuscript. We will improve the readability in methodology section to strengthen the conceptual set-up.

**Major Comments:**

The main approach is to use Convex-Hull Non-negative Matrix Factorization (CH-NMF) to infer possible end-member compositions by searching for a simplex that optimally encloses the stream water observations. The assumption for this is, based on authors, that end-members are located near the most extreme points that bound the observations in "mixing" space. From this assumption, it is clear that a simplex is basically determined by the data structure of observations, in other words, the shape of the sample cloud. What if one or more extreme points are missing in our observations? This could happen if samples are collected sparsely or only on certain hydrologic conditions/seasons that do not contain extreme samples (samples with extreme concentrations for at least one solute). The number of samples could also change how samples are distributed. With the same data set, can similar results (with reasonable uncertainties) be obtained from subsets of samples with varying number of samples that are randomly selected?

> Thank you for your insightful comment. This is indeed a drawback, and we have mentioned briefly in the manuscript that CHEMMA can only identify end-members that are well-sampled in the data. We will expand on this point in the revised manuscript by highlighting this issue in both abstract and introduction. We acknowledge that some of the fundamental assumptions could limit the CHEMMA application. Improvements that overcome these limitations are left in future work.

There is a lack of conceptual setup where this study came from and where it goes in relation to existing tools in EMMA, particularly the diagnostic tools of mixing models (DTMM; Hooper, WRR, 2003). In one study, Christopherson and Hooper (WRR, 1992) specifically concluded that "Unambiguous identification of the source solution compositions from the mixture alone is impossible; thus, it is necessary that potential source solutions be derived from independent measurements." I do not mean this conclusion cannot be challenged, but the rationale must be stated clearly and explicitly, possibly using a conceptual set-up. Also, what is its relation with DTMM? Will the current study be supplemental or a substitute to DTMM in regard to the number of end-members? Can DTMM actually help to enhance CHEMMA and how?

The study used data collected in late 1980s. That is okay but what I am concerned is about the conservativity of all six solutes. How can we be convinced if all six solutes are conservative? If any of those is not conservative, the results of CHEMMA would be different. In my opinion, this is where DTMM may be able to help. Also, isn't it interesting to compare the number of end-members acquired using CHEMMA to DTMM?

We appreciate your suggestion on improving the understanding of practicing CHEMMA. We agree that the conceptual set-up is not clearly stated in the paper. In the revision, we will modify the last paragraph of the introduction (l55 – l65). We have added a sentence in l65 to clarify the conceptual setup:

*"Christophersen and Hooper (1992) suggested that "[u]nambiguous identification of the source solution compositions from the mixture alone is impossible". In a strict sense this is likely true, since the underlying assumption (streamflow as a conservative mixture of invariant sources) is unlikely to be adhered to in a real watershed. However, we believe there may be utility in developing tools that can seek some insights (perhaps not free of ambiguity) into the potential source solution composition from the observed mixture. We propose CHEMMA as an attempt to push this boundary and to see how far we can get."*

As we said to Referee #1's comment, we have added a paragraph after l200, and discussed DTMM in point 4:

*"For most hydrologists, end-member analysis is used to identify the water sources, and toward that purpose CHEMMA may be useful in the following ways: 1. CHEMMA may be used to reduce subjectivity when selecting from field-measured end-member candidates by comparing them to CHEMMA end-members; 2. CHEMMA may identify end-members that have not been sampled in the field, which may serve as a check for missing sources; 3. CHEMMA end-member compositions may help hydrologists ask better questions and provide guidance for field sampling by suggesting source characteristics; 4. CHEMMA can be used in conjunction with the Diagnostic Tool of Mixing Models (DTMM, developed by Hooper (2003)). DTMM is used to assess the tracer conservation, and mixture rank. CHEMMA can be enhanced by using DTMM analysis to select conserved tracers for analysis. The robustness of CHEMMA end-members also serve as a check for DTMM-determined rank of mixture."*

**Minor Comments:**

L18: Before the first reference, add "e.g.,". Many classical references on EMMA were not actually cited.

Thank you for this suggestion. We have added "e.g.", and also added two new references mentioned in your comment (Liu et al., 2008 and 2017) as applications of EMMA under different climatic settings.

L24: This statement should refer to conservative solutes.

Thank you for bringing up this confusion. We added a word "solute" between "chemical composition" to clarify that the sentence is talking about solute conservation.

L30: "Streamwater concentration are naturally correlated." It is true if you refer to conservative solutes; otherwise it is an ill statement. Use two words "stream water" instead of one word "streamwater". Also, use plural for "concentration".

Thank you for your suggestion. We changed "streamwater" to "stream water" for this manuscript. We also adopted Referee #1's suggestion and changed this sentence to:

*"Stream water concentrations of different conservative solutes tend to be correlated."*

L28: The second one is no longer a hypothesis or assumption because of the diagnostic tools of mixing models by Hooper (2003); See Liu et al. (WRR, 2008) for a demonstration and how

this was addressed.

L45: True traditionally but not after DTMM is developed. See Liu et al. (WRR, 2008, 2017) as examples.

L51-52: Not true with DTMM.

L52-53: True but DTMM can help identify conservative solutes so that users can use only conservative ones. I mention this because I think your study is also based on mixing of conservative solutes. This should be stated/defined earlier in your text.

L186-206: Need to indicate where this modeling will lead to and how it may work together with DTMM.

**We would like to response to these five comments collectively. Thank you for your recommendation about DTMM and related application papers. We agree that DTMM workflow is a good complement to both EMMA and CHEMMA. And we added a paragraph to clarify how DTMM and CHEMMA can potentially work together. Please refer to the response to the last major comment above for more details.**

L33-35: Multiple issues here. (1) Is Pobs actually eigenvectors? If so, use a parenthesis to annotate so; otherwise explain what it is and how to calculate it. (2) Get rid of the redundant "the". (3) My understanding is that once a standardized data set is used, a correlation matrix is decomposed rather than covariance matrix. Check if this is correct.

**Thank you for carefully checking the mathematical details. The rows of $P_{obs}$ are the eigenvectors of the correlation/covariance matrix $X_{obs}$. We have added a parenthesis segment: *(rows of which are eigenvectors of the correlation matrix)*, and we have deleted the redundant "the" appearing later. Because $X_{obs}$ is standardized observation, the correlation matrix and the covariance matrix are essential the same. Performing eigendecomposition on both matrices yields the same results. We have adopted your comment to change the covariance matrix to correlation matrix to make it clear.**

L36: If P are indeed eigenvectors, cite Christopherson and Hooper (1992) for the equation.

**Thank you. P are eigenvectors. We have cited Christophersen and Hooper (1992).**

Result 2: Eigenvectors and PCs are different. PCs are calculated based on eigenvectors and observed concentrations.

**Thank you. We adopted a terminology in this manuscript consistent with usage in applied mathematics literature, such as Jolliffe (2002). In our understanding, eigenvectors are derived from the correlation matrix of the observed concentrations by performing eigendecomposition (as used for this manuscript) or singular value decomposition. Resulting eigenvectors are orthogonal bases as known as Principal Components (PCs) (Jolliffe, 2002). Loadings are the coefficient calculated based on eigenvectors (PCs) and observed concentrations (Jolliffe, 2002), and are referred as contributions (of end-members) in this manuscript.**

L93: I still think it is correlation matrix not covariance matrix. Also, what you mean here is eigenvectors not PCs.

**Thank you. As we responded before, in the revised manuscript we have changed covariance matrix to correlation matrix for clarity. We used eigenvectors and PCs**

**interchangeably according to our reference of PCA terminology (Jolliffe, 2002).**

Result 3: Is it specified anywhere how to project mathematically?

**Thank you for finding this confusing part. Projecting a matrix A to another space through a projection matrix $P^T$ to get projected matrix B is defined as $B = AP^T$, just as Equation 1 and 2 show. We added a parenthesis fragment:** *(similar form as Eqn. 1 & 2).*

Result 4: Will the dimension of S differs from one projection plane to another?

**Thank you for noticing this technical detail. Yes. S records all boundary points in each projection plane and the number of recorded points at each plane can be different.**

Result 5: Is X expression actually [[xem1], [xem2], . . ., [xemk]], as each xemi has a dimension of n by 1?

**Thank you for noticing the dimension consistency. Yes, $x_{emi}$ has dimension of n by 1. We have checked the consistency of dimensions in Algorithm 1 a couple of time before submitted the manuscript.**

L125: I think "equifinality" is part of your talking here. Why not citing "equifinality" directly? It is a common term that hydrologists are very familiar with.

**Thank you for your comment. The concept here is slightly different from equifinality. This paragraph particularly talked about limitation of an optimization problem on minimizing the objective function in Step/Result 5.**

**Thank you for pointing out style problems. We have adopted the following comments.**

L31: Need at least one reference (e.g., Christopherson and Hooper, 1992).

L41-42: Cite Hooper (WRR, 2003).

L60: Need to specify "extreme points". I think you refer to "extreme points of stream water samples".

L64: I think you mean "end-members' composition".

L94: Spell out PCA as it appears for the first time.

L102: Specify the constraints, each between 0 and 1 with sum of all to be 1.

**References**

**Liu, F., Bales, R. C., Conklin, M. H., & Conrad, M. E. (2008). Streamflow generation from snowmelt in semi-arid, seasonally snow-covered, forested catchments, Valles**

Caldera, New Mexico. Water Resources Research, 44(12).

Liu, F., Conklin, M. H., & Shaw, G. D. (2017). Insights into hydrologic and hydrochemical processes based on concentration-discharge and end-member mixing analyses in the mid-Merced River Basin, Sierra Nevada, California. Water Resources Research, 53(1), 832-850.

Jolliffe, I. T. (2002). Mathematical and Statistical Properties of Population Principal Components. In Principal Component Analysis (pp. 8-22). Springer, New York, NY. https://doi.org/10.1007/b98835

---

## Author Response (AR2)

hess-2020-250 Submitted on 24 May 2020

**A data-driven method for estimating the composition of end-members from stream water chemistry observations**

Esther Xu Fei and Ciaran Joseph Harman

**Response to Reviewers**

We would like to thank the editor and reviewers for the useful suggestions, which have improved the manuscript. In particular, we have included several major additions:

- Greatly expanded analysis of the limitations, robustness, and uncertainty of the proposed approach, including:
  - Analysis of the uncertainty in end-member identification using bootstrapping
  - Analysis of the effect of dataset size by subsampling the data
  - Analysis of how data structure controls the robustness of the parameter identification using synthetic datasets
- *Expanded discussion of the relationship between the proposed method and previous literature*
- Discussion of the relationship between CHEMMA and DTMM

These additions have necessitated resubmitting the paper as a regular research paper rather than a Technical Note. We look forward to further feedback from reviewers. Below we provide responses to the second round of reviewer comments on the technical note.

**Reviewer #1**

The authors proposed an interesting method to decompose stream water into end-members using stream tracer data alone. I like this idea very much.

**Thank you for this encouragement, and for the helpful suggestions.**

However, the shortages of the method are apparent. For example,

 it relies heavily on the used data. When the used stream samples are different, the method likely yields different identifications of end-members. The numbers of stream samples, the extreme points involved in the input data, the tracers measured from the stream samples, as well as the seasons during which the stream water were collected, have significant impacts on the outputs;

We agree that this weakness was not adequately addressed in our previous version of the paper. In the new version we have implemented two additional modeling experiments. First, to examine the influence of sampling variability we bootstrapped the original stream water samples 1000 times to estimate the resulting variability in identified end members. Second, in order to understand the effect of the numbers of stream samples, we also sub-sampled the original dataset with smaller sample sizes. The results of these analyses can be found in section 3.3 Uncertainty analysis from line 253 to 280, and figures 5 and 6.

- (2) The interactions of the tracer concentrations and contributions to stream water of endmembers were not treated well. An end-member with extremely high or low tracer concentration may not necessarily result in extreme concentration of stream water when its contribution is rather low. I am wondering the ability of the current method to identify end-members with low contributions to stream water.
- (3) Seems extreme points in the data series of stream tracer concentration are required for the implement of the method. If the collected stream water samples do not show any outliers, does the method still work? The mixture of end-members with distinguished tracer concentrations not certainly result in much changes in the stream water tracer concentration, considering their contributions are changing in the time periods.

**Thank you for identifying these important points. We have included new analysis and discussion that addresses both. This includes further analysis of the Panola results, and analysis of a set of synthetic datasets.**

We should note that the end-members identified using our method do not necessarily have extreme concentrations. Instead they have "unusual" concentration profiles when all variables characterizing them are viewed collectively. That is, they are at the 'fringes' of the data cloud in principle component space. For example, in the revised manuscript, this can be seen in the newly added Figure 5, where the predicted hillslope end-member does not have very extreme SO4 concentration, but rather was identified due to its uniqueness in other dimensions compared with the other two end-members. This point is discussed in the manuscript at line 259:

"Figure 5 also re-emphasizes that CHEMMA identifies end-members that exhibit collectively unusual combinations of concentrations (i.e., vertex-like structures in the overall data cloud). While many solute concentrations of CHEMMA predicted end-members are located towards extremal values of the observations, they need not be all individually extremes (e.g. the sulfate concentration of end-member 3, corresponding to the hillslope end-member, Figure 5 upper middle plot)."

However the end-member identification has less to do with the extremes than it does the overall boundary of the dataset in n-dimensional space. In other words, it matters more that there are some samples in which an end-member is absent than that there are some where it dominates. You are correct that if a source consistently contributes a minor fraction of the total, it would not be identified as an end-member, since its unique profile would not have an effect on the dataset boundary. However, this dependence on the overall boundary means that relatively few samples may be necessary to robustly identify an end member. We examined this question by generating synthetic datasets with varying degrees of imposed 'end-member' structure. These were generated by creating normally distributed random concentration profiles (essentially spheroids in n-dimensional concentration space), and then censoring samples that fell outside

the mixing space. By varying the variance of the distribution used to generate the samples we can generate datasets influenced to varying degrees by the end-members. The results show that end members can be robustly identified (albeit with some systematic bias, namely an underestimation of their 'extremeness') when only <1% of the synthetic data contained samples that entirely lacked at least one end-member. This point is discussed further in the manuscript in section 3.4 and in figure 8 and 9.

CHEMMA does not require outliers but the structure of the data points. To demonstrate this, we have added a synthetic data section to express when CHEMMA fails and when it starts to work perfectly (Figure 8).

(4) The method may not be able to identify end-members with similar tracer concentrations. This may be not important when we are focusing on the components contributing to the tracer concentration of stream water. But identifying runoff components with similar tracer composition could be very important to understand the changes of hydrological processes in catchments.

We acknowledge this is a problem – however, it is not a problem that is unique to CHEMMA. Nevertheless, as demonstrated in previous section two source waters with similar tracer concentrations may be distinguishable by CHEMMA if enough samples are available in which one is absent and the other dominant. This is discussed on Line 259 (given above) and Line 315:

"The CHEMMA algorithm appears to detects structure more robustly when the dataset includes samples containing very small contributions at some time. However, a consistently very low contributed end-member will not be effectively detected because it does not affect the shape of the data cloud boundary."

**Reviewer #2**

After I examined the revision, my concerns about the validity of this pure mathematical tool grow instead of diminish. This mathematical tool is surely beautiful, but whether or not it yields hydrochemically meaningful results has not been well demonstrated. No matter how beautifully can chemical concentrations in end-members be inversely derived mathematically from streamflow chemistry alone, it has to be proved to be accurate with foreseeable and acceptable uncertainties. The uncertainty I am talking about here is not the uncertainty arising from chemical analysis as we always know but one caused by this tool per se. There are two questions that are related to this type of uncertainty, which were inquired earlier but not actually addressed in the revision. If the number of samples changes significantly, can chemical concentrations in end-members be still determined within an acceptable range? If non-conservative solutes are included in the analysis, are the results of chemical concentrations in end-members and the number of end-members determined by this tool consistent and valid? These questions have to be answered with actual data analyses before it becomes a convincing tool. As it currently stands, I do not feel it is ready for others to use this tool with high confidence.

Chemical concentrations in end-members were determined by data structure by CHEMMA. No doubt that a significant change in the number of samples will change chemical concentrations in end-members. But as long as the determined concentrations are within certain range, we are fine with it. Simply saying CHEMMA applies to large data set without a demonstration is not an appropriate answer. Also, how large can we consider it to be "large"?

We appreciate these suggestions on how to improve this manuscript, and have made extensive additions to the manuscript to address them. We have added sections to examine how the robustness of the identification depends on number of samples available, the uncertainties associated with the sampling variability, and the robustness of the method to different data cloud structures. We may not be able to develop a universal standard to quantify how many samples is 'enough', this really depends on the structure of each given study dataset. However, as we saw from the results of exploring the Panola dataset (Figure 6), CHEMMA converged on some components of some end members with as few as 45 samples, while for others as many as 500 samples were needed before the estimated component concentrations converged to the values estimated using the full dataset of 905 samples.

In my opinion, non-conservative solutes should not be included in the analysis, nor can chemical concentrations of non-conservative solutes in end-members be derived from CHEMMA. Otherwise, CHEMMA should be a totally different set-up and have to deal with chemical equilibrium. With non-conservative solutes included in CHEMMA, however, solutions can be achieved mathematically by increasing the number of end-members. Rick Hooper noticed this problem in EMMA in his later work, which eventually led the publication of diagnostic tools of mixing models in 2003. Including non-conservative solutes in EMMA will cause polygon to bend outward. Yes, this problem can be mathematically resolved by adding additional "endmembers" to obtain a more complex polygon, but they are not truly end-members because in such cases the assumptions of mixing models are violated. Analogically, the same issue applies to CHEMMA. Why not testing using DTMM if all solutes included in CHEMMA are conservative and examining whether or not non-conservative solutes caused the fourth endmember? Also, why not determining the number of end-members using DTMM and comparing with CHEMMA? Simply citing the published results of Hooper et al. (1990) cannot guarantee those solutes are conservative, as conservative behavior of those solutes, along with the number of end-members, were determined subjectively at the time.

We acknowledge your thoughtful comment, and agree that non-conservative solutes should not be included. CHEMMA is based on the assumption of conservative mixing, as is EMMA. We agree that DTMM should be used conjunctions with CHEMMA to test for whether solutes are conservative, and to select the number of end members. CHEMMA is compatible with DTMM in this regard. We have introduced a new section discussing the use of DTMM for these purposes. In particular:

"To carry the idea of DTMM rank determination further, we performed a fivefold cross validation analysis on PCA fit residuals on the Panola data with varying dimensionality (Figure 4). The mean square errors of residuals (Figure 4a) exhibited the greatest decrease when the dimension was increased from one to two, which suggests three end-members might constitute a parsimonious set. However, the small normality test p-value in Figure 4 b) shows that residuals of sulfate, magnesium, and calcium solutes still maintain some structures in a two dimensional mixing space. Residual structures persist until the dimension goes beyond five(Figure 4 b)). Thus even with DTMM, the 'true' rank of the dataset remains uncertain. However, DTMM analysis at least provides an established method to identify conserved solutes and to justify the choice of rank. The robustness of CHEMMA end-members could also serve as a check for DTMMdetermined rank of mixture."

**Reviewer #3**

The authors present a promising method that allows to identify and characterize end-members using stream water tracer concentrations only. While I believe their work is a valuable contribution to existing literature, one major shortcoming is the lacking literature review and thus relating their work to the existing literature (beyond EMMA). I am aware that the field of end-member mixing modeling is wide, however, the authors miss to acknowledge key papers of the field of hydrology such as Carrera et al., 2003, (https://doi.org/10.1029/2003WR002263) or Genereux, 1998 (https://doi.org/10.1029/98WR00010). Likewise, they neglect recently published papers that provide methodological advances in mixing analyses within hydrology, e.g., Beria et al., 2020 (https://doi.org/10.1029/2019WR025677), or Barbeta and Peñuelas, 2017, (https://doi.org/10.1038/s41598-017-09643-x). The authors claim that no method exists to characterize missing or unmeasured end-members, however, Popp et al., 2019, have provided an approach that allows to identify unmeasured end-members.

**Thank you for identifying these key papers. The cited papers focus on similar topics but take different approaches (using Bayesian framework to reduce and quantify uncertainties) than we are presenting. We have added a paragraph in the introduction to acknowledge their work.**

"It is worth distinguishing CHEMMA from previous applications of statistical learning methods (such as maximum likelihood estimation, Bayesian inference, and Markov Chain Monte Carlo, MCMC) to end-member mixing analysis. Genereux (1998) presented a linear estimator for uncertainties in end-member concentration and mixing ratios. Carrera et al. (2004) achieved something similar using a maximum likelihood method. By combining likelihood methods, Bayesian inferences, and probabilistic linear models with a MCMC algorithm, various authos including Barbeta and Peñuelas (2017); Beria et al. (2020); Delsman et al. (2013); Popp et al. (2019) have been able to obtain time-evolving uncertainty estimates. These contributions all focus on quantifying uncertainty resulting from the use of field-sampled candidate end-members. In contrast, CHEMMA aims to infer the end-members themselves." Popp et al., 2019 does provide an approach that allows identification of unmeasured endmembers, but we would also like to clarify the difference between our approach and theirs. Popp et al., 2019 introduce a single residual end-member that represents the collective effect of all unobserved end-members in addition to the observed ones, which are used to initiate the Bayesian mixing model. That is, their approach is best suited to where information on some observed end-members is available. However, CHEMMA do not require any observed endmembers. It relies solely on the mixture data. The contrast with the Popp model is discussed on line 79:

"Popp et al. (2019) comes close, introducing a residual end-member that represents collective behavior of all unobserved end-members. Their method still requires some a-priori knowledge of "observed" end-members to initialize a Bayesian mixing model. In contrast, CHEMMA allows for inference of the entire suite of end-member compositions, and their associated uncertainties."

Another shortcoming is that the method is only applied to one data set. I strongly suggest to apply the method to other data sets (using different tracers and in a different geographic setting) to prove the robustness of the method. It's been shown that the tracer set size has a major influence on end-member mixing modeling (Barthold et al.). Testing the method on other datasets should be feasible since many datasets are readily available nowadays.

It is certainly true that the validity of the approach will be strengthened by applying it to multiple datasets. We have chosen not to here, and hope that the material we have included in the new manuscript is sufficient to motivate further investigation. The reviewer's point is well taken though, and we have partially addressed it with the addition of the synthetic datasets in section 3.4, which we use to analyze to examine the robustness of CHEMMA. We have also added a section where we subsample and bootstrap the Panola dataset in order to explore sources of uncertainty and the robustness of the results. We look forward to applying CHEMMA on other datasets in the near future.

I really appreciate the detailed description of the methods and the valuable reflections (section 4) added to the latest version (v3). It is also great that the code and data are available in a Jupyter notebook.

**Thank you for recognizing our work in this area. We will continue refining the code and may make it to be a Python package in the future.**

The manuscript is well written and clearly structured, however, readability should be improved by correcting for a couple of grammatical flaws (e.g., articles are often missing) that I believe to have detected.

- L. 4 and throughout the text: consider replacing "candidate" with "potential"
- L. 24: replace "should" with "can" be explained
- L. 28: the hypotheses

• L. 29: I would rephrase it saying that the "1) stream water consists of the identified endmembers and 2) all end-members were identified correctly".

• L. 63: please add a reference to the 4th statement

• L. 67-68: this statement is not true. See comment above about method provided by Popp et al., 2019, (https://doi.org/10.1029/2019WR025677)

• L. 74-76: I would really appreciate to see this method applied to other datasets

• L 92: "end-member mixing" approach/method?

• L. 96: subspace

• L. 152: analysis many times (instead of "a large number of")

• L. 163: can you please specify "reasonable"?

• L. 181 following: please statistically quantify the similarity between field measurements and your values. E.g., not only alkalinity (hillslope) seems to differ considerably. Also, consider removing decimal

points in Table 1 given the high st.dev.

• L. 204: please use a statistical test to quantitatively describe how well you can reproduce endmembers of Hooper et al.

• L. 225: that is a great suggestion! You could also indicate that a time lag representing a delay caused by tracer transport from the source to the output (see Beria et al.) adds uncertainty.

• L. 232: I would remove this statement.

• increase font size in Fig. 3

• Figs. 2 and 3 and table 2: specify what is meant by "organic"

Thank you for helping in improving the readability of our manuscript. We have adapted your comments in our manuscript.

---

## Author Response (AR3)

*A data-driven method for estimating the composition of end-members from stream water chemistry timeseries*

*by Esther Xu Fei, Ciaran J. Harman*

**Response to reviewers**

September 20th 2021

**We would like to thank the editor and reviewer for their thoughtful reviews, and for their patience as we have revised this paper. Below we respond to the individual concerns expressed by the reviewer**

Major Concerns

Conservative behavior of tracers used in the analysis was not tested. I had this concern before, but it was not addressed adequately or correctly in the current revision. I suggested to run DTMM first to determine the conservative behavior of all solutes (e.g., completely conservative, semi- or quasi-conservative, and non-conservative under a lower dimensionality) and then to include conservative ones in CHEMMA. Instead, authors ran DTMM after CHEMMA and just compared the outcomes. They found that the residuals of sulfate, magnesium, and calcium still maintain some structures in a two-dimensional mixing space and the residual structures persist until the dimension goes beyond five (Figure 4 b). If this result is true (I am not sure if Shapiro-Wilk test is appropriate for this analysis; see my comment on Figure 4 below), it strongly indicates some solutes are not completely conservative. With six solutes, you cannot go beyond five dimensions to determine conservative solutes and the number of end-members (see my comment in the second round of revision). Authors misunderstood how DTMM works.

**Thank you for these suggestions. It is important to note that the data used in the paper is a subset of precisely the same data (Panola Lower Gauge) as was used in the original DTMM paper (Hooper 2003). In other words, DTMM has already been applied to this dataset and the results published. Figure 7 in Hooper (2003) presents the concentration relation in the observational space, and Figure 9 in Hooper (2003) presents the residuals against concentration in one-, two-, and three-dimensional mixing space. Hooper (2003) does not conclude any solutes should be excluded from the analysis due to non-conservative mixing. Instead, he suggests four end-members (three-dimensional mixing space) may be warranted, which is consistent with the results presented here.**

**Given that our results are entirely consistent with the paper that first presented DTMM, and make use of the same dataset, we are unsure how to satisfy the concern of the reviewer. This issue also seems somewhat peripheral to the main contribution of the paper, which is to present the CHEMMA method. Certainly the choice of an appropriate number of dimensions is important, but it is a challenge for EMMA, CHEMMA, and all data**

**analytic techniques that rely on low-dimensional embeddings of high dimensional data. It is not the intention of this paper to solve that particular problem, and it seems unreasonable that the paper might not be accepted because it fails to due so.**

With that said, I do not mean you cannot run CHEMMA with all six solutes together or with different groups of solutes. Instead, I do encourage authors to run different versions of CHEMMA with various combinations of solutes (following the DTMM results) and compare the outcomes, including the number of end-members.

**While we appreciate the suggestion, we do not believe extending the length of the paper in this way would add anything of value. As we discuss above, there is no reason to exclude any of the solutes and so we do not see the point of running different combinations of them. The goal of this paper is to present the CHEMMA method, not to examine the influence of using different set of solutes.**

As a research article, I strongly suggest to run DTMM first (following Hooper 2003), then EMMA (similar to Christophersen and Hooper 1992), and finally compare with the results of CHEMMA (e.g., groups of all six solutes and conservative solutes identified by DTMM as mentioned above). The comparison should not be limited to the end-member composition, but include the number of end-members, the fractional contributions of end-members, and the end-member distances. For example, how do the end-member distances from the end-member composition of CHEMMA compare to that of the measured ones? Is there an improvement in the end-member distances with CHEMMA? I did not keep track of whether or not Hooper (2003) used exactly the same data set as Hooper (1990). If so, you may not need to re-run both DTMM and EMMA but just summarize their results.

**We thank the reviewer for these detailed suggestions. Many of these are already addressed in Christophersen and Hooper (1992), Hooper (2003), and in the present paper. End member distances for both EMMA and CHEMMA are shown in figure 3. The change in the fractional contributions of each end member can also be readily inferred from this figure (e.g. the fraction of water supplied by the CHEMMA end-member most like the organic end-member from Christophersen and Hooper (1992) is less than that inferred in the original study.**

**Beynd that, it is unclear to us what further application of EMMA and DTMM to this dataset will produce – certainly we cannot see how it better achieves the aims of the paper.**

Addition of a test with varying sample sizes (Figure 6) is nice and very much appreciated. One result is very much promising (e.g., relatively stable compositions for end-members 2 and 3), but others are not (e.g., significant variation of composition of end-member 1; still many outliers for end-members 2 and 3). Together with significant variability in identifying end-members of the synthetic data (Figure 8) and high uncertainties of algorithm (Figure 7), it indicates that the data

structure or the distribution of sample points determines the end-member composition. The role of CHEMMA in end-member mixing analysis is limited. This limitation should be explicitly discussed and stated in the abstract and conclusion. This does not downplay CHEMMA's value, but simply tell the truth so that future users will not be misled. As a matter of fact, CHEMMA would be very helpful in identifying a missing end-member, guiding field sampling of end-members, and generating a hypothesis test.

**We appreciate your sincere suggestion, and we have added clarifications in both abstract and conclusion. In Abstract, we changed the part after Line 12 to:**

**"We examine uncertainties in end-member identification arising from the random initialization of the CH-NMF algorithm, from the sample size, and from the data structure using both real and synthetic data. The results suggest that the robustness of the CHEMMA method depends on the dataset including, for each end-member, a subset of samples in which is nearly absent, as well as others in which it is a more substantial part."**

**We also added to the Conclusion at Line 355:**

**"However, the usefulness of CHEMMA is limited by the structure of the data in mixing space. As Figure 8 suggests, CHEMMA will fail for datasets in which all end members are present in all samples to some non-trivial degree. Samples in which an end-member is absent provide critical information, and strongly control the location of the face of the convex hull used to identify the other end members."**

Moderate Comments/Concerns

Abstract:

The manuscript has been modified, with a new section (re: synthetic dataset) and a new analysis (re: varying number of samples) added, but the abstract was not updated to include the results from these analyses, nor was the limitation of the approach stated in the abstract.

**Thank you. We have updated the abstract.**

Introduction:

Somewhere in the introduction, all of the mixing model assumptions needs to be explicitly listed (i.e., i. Tracers used in the mixing model must be conservative; ii. The number of end-members is known; iii. End-member compositions must be distinct for at least one tracer; iv. End-member compositions are spatiotemporally constant or their variations are known or treated as different end-members). These assumptions should be discussed, e.g., which ones have been addressed by which tools and which ones are still up for research. In my opinion, the first two assumptions have been resolved by the diagnostic tools of mixing models, which has to be acknowledged to respect the earlier study. This will pave a clear pathway for your own research. However, this does not mean DTMM cannot be challenged or improved.

**Thank you for your suggestions, we have added these assumptions to the end of the second paragraph in the Introduction.**

**"The EMMA method assumes that 1) solutes used in the mixing model are approximately conservative, 2) stream water consists of identifiable number of end-member sources, 3) end-member compositions are distinct for at least one tracer, and 4) end-member compositions are spatiotemporally constant (or their variations are known or can be approximated using additional end-members)."**

Saying that including non-conservative solutes in the mixing models has not been resolved is inappropriate and misleading, as non-conservative solutes should not be included in the analysis based on the mixing model assumptions. This does not mean you cannot challenge the assumptions, but I do not think that is what your study aimed at.

**Thank you for your suggestions, we have added a sentence after 3) in Line 73 that "therefore, only tracers that are believed to be approximately conservative (i.e. since they entered the stream their concentrations have been altered primarily by dilution rather than other mechanisms) should be included."**

Results:

How do the fractional contributions compare between your and Hooper's results? Were the fractional contributions of the fourth and fifth end-members significant compared to the three end-members used by Hooper? How do the end-member distances change with the end-member composition of CHEMMA?

**As we stated before, we have limited our analysis to comparing the similarity of the CHEMMA and Hooper end-members.**

Figure 4: Is the Shapiro-Wilk test appropriate for this analysis, as it is usually used for normal distribution test?

**We agree that the SW test is not an appropriate check on the result, as it only tests whether the overall residual distribution is normal, and not whether there are higher-order relationships between residuals and the independent variables. References to this test and the associated figure have been removed.**

Miscellaneous Comments and Suggested Edits

P1/L2-3: This statement is not completely true, not exactly part of the mixing model assumption (see one of my comments above). As long as the temporal/spatial variation is known, a hydrograph separation is still valid.

**We have deleted the inaccurate phrase "relatively time invariant".**

P1/L17-18: This is not exactly true. The end-member composition does not have to be constant. Delete the phrase.

**We have added a word "unknown" to specify the question that we are trying to answer.**

P2/L26: Again, it does not have to be temporally invariant.

**We have deleted the inaccurate phrases.**

P3/L63-65: This is not what Hooper (2003) means. Instead, he demonstrated the filature of using the rule of one. He suggested to use residual distribution pattern. Indeed, their 1992 paper followed the rule of one (as you stated in the sentence following this one). You have to follow the temporal evolution of EMMA and cannot use an early one to reject a later one.

**We have changed the order of the statements to better follow the actual timeline.**

P3/L69-74: With DTMM, the #2 is not true. The #3 should not be stated as mixing is subject to conservative tracers and so is CHEMMA as I talked above. I do not think CHEMMA can include non-conservative solutes.

**We have added a sentence to the end of #2 and changed #2) to "determining the number of significant PC is subjective to some degree, even with the aid of DTMM".**

P4/L113: Why not running DTMM before CHEMMA?

**DTMM has been applied to this dataset, and the results are discussed in Hooper (2003).**

P4/L114: Based on Figure 1, "…projected into the 2D subspace spanned by pair-wise PCs"?

**We have added a mathematical expression as an example to illustrate this point to avoid repeating.**

P5-6: After reading the text, it is still not clear how both I- and J-vectors were generated. Through an optimization procedure itself?

**We have added a section after Line 127 to reflect this comment:**

**"Because SI may be any possible points within the convex-hull constructed using S, J is needed to force SI to be chosen close to the convex-hull boundary. In practice, I and J are estimated iteratively using an optimization procedure until they converge (Eq. 1 and 2 from Thurau (2011)."**

P6/L158-163: Kind of arguments are needed to set the stage for multiple runs. But the exact statements here fit better to discussion.

**The paragraph has been updated to better communicate the objective.**

P7/L187: You cannot just cite Hooper et al. (1990). DTMM has to run to identify conservative solutes and the number of end-members.

**We added a sentence at Line 191 to reflect this comment:**

**"Later Hooper (2003) suggested the rank of the data (Lower Gauge in Hooper (2003) dataset) is at least 3."**

P9/L253-259: Some of them are very much speculative.

**Speculative parts of this paragraph have been reduced, so it now reads:**

**"Failure to conform to these assumptions undermines the validity of the method. For example, rare contributions from an end-member can result in the the dispersion of Cluster 3 in Figure \ref{fig: comparisons}b. Temporal variations of the end-member composition could produce the kind of variations seen in PC 3 in Figure \ref{fig: comparisons}d . Fortunately, CHEMMA itself may be a basis for exploring the effects of time-variability. For example, by partitioning the dataset into time periods (or hydrologic state, etc), the apparent temporal variability of end-members could be explored."**

P11/L325-330: These are not conclusion.

**This may be a matter of style, but we find it useful to incorporate a brief summary in the conclusions.**

Title: Get rid of "observations", which I think is redundant.

**Observations has been replaced with "timeseries".**

P1/L1: Delete ", and is".

**Done**

P1/L4: Change "additional measurements" to "samples".

**Done**

P1/L5 and also L12: I think this article is no longer a technical note.

**Done**

P1/L16: Delete "profile".

**Done**

P2/L26: Change "observations' to "samples".

**Done**

P2/L28-30: This is probably where you state all the assumptions of mixing models as I suggested earlier.

**Done**

P2/L33-35: Any citation(s) for this statement? This is the most important statement to justify your study.

**Done**

P2/L40: Add "estimate uncertainties of" after "to" if those are true.

**Done**

P3/L61-63: This is only one of a few criteria used to screen end-members (see Hooper, 2003; Liu et al., 2008 and 2020).

**Done**

P3/L81-82: Change "allows for identification of" to "aims at identifying", as by far you have not yet demonstrated if you can.

**Done**

P3/L84-85: I think what you want to say here is that end-member composition does not have to be distinct for all tracers (assumption iii above).

**Done**

P4/L95: Change "find" to "determine".

**Done**

P4/L115: Add "at each 2D subspace" at the end of the sentence. Then, get rid of the middle sentence if "pair-wise" is added as suggested above.

**Done**

P5/L (not clear which line but the statement following "Result"): Does x-matrix contain the standardized values or original concentrations? Need to specify.

**Done**

P5/L (#4 in the table): Change "needed" to "found" because the number is variable and it does not matter for how many to be found.

**Done**

P5/L (#5 in the table): I am sure both I-vector and J-vector were explained in the text, but in the table their function still needs to be specified so that readers understand what SI-matrix means.

**Done**

P5/L (#6 in the table): Need to say h and H represent for fractional contributions.

**Done**

P7-8/L193-215: Most if not all of them should be presented under 3.1.

**Thank you for helping us improve the quality of illustration. We answer the changes of the following questions collectively.**

Figure 1: What are the red crosses in c? Why are they in different positions from those in b? Also, need to say this uses 3D as an example.

**We added a sentence at the end of this caption that "The red crosses are the same extreme points marked in b), but are projected back in the three dimensional PC space." And we added a part behind the first sentence that "using a three dimensional space with four end-members".**

Figure 2: Do not use the same colors for the PC axis's of solutes as for the triangles; way too confusing. You used "diamonds" not "squares". In the case of four end-members, this is not exactly the convex hull but the projected convex hull into 2D PC subspace. For four end-members, it needs three PC subspaces. This should be specified.

**We have changed the color for the end-member diamonds to make the visualization clear. We have also corrected the name for the markers and added a sentence that "Note that three dimensional subspace is required for four end-members" at the end of the caption.**

Figure 3: Do the colors shown in legend match those points on the plot? I do not see blue and red colors in the legend. Also, the same legend used in all cases regardless of the number of

endmembers. Is it necessary to show all four (maybe five) when you are seeking for only three endmembers?

**The reason why those colors seems to not match with the legend is that we used transparent markers, where "the color shade of each cluster reflects the concentration of the vertices at its location". And we added the quoted sentence to the end of the caption.**

Figure 4: Cannot be self-explained. Averaged from 100 runs? Residuals of what? What does five-fold cross validation mean? Font size too small; resolution too low; symbol size too small too. Very hard to distinguish Ca and Si curves.

**Thank you for pointing out this confusing phrase. We have changed the "Average scalar measures" to "Scalar measures", and increased the font size.**

Figure 5: Observed in both stream water and end-members? Do not use "observation" for "stream water" as observed end-members were also shown.

**We have changed the original "Observation" x-axis tick to "Stream water". We also modified the caption to make it explain better.**

Figure 6: You mean "the same size ..."? Also, change "sample" to "samples". Font size too small.

**We have changed the original "Observation" x-axis tick to "Stream water". We also modified the caption to make it explain better.**

Figure 7: How were algorithm and data uncertainties calculated? I do not think they were covered in the text.

**We have added the original "Observation" x-axis tick to "Stream water". We also modified the caption to make it clearer.**

Figure 8: (top) Different number of samples? Or, different sample distributions? (caption) Do they all have the same number of samples?

**They have the same number of samples and distribution prior to truncation to the mixing space. We have added a sentence in the caption to demonstrate the same sample size and distribution that "All cases (1 to 6) have the same number of samples (1000 samples) and are normally distributed around the inner center of the grey triangle."**

Figure 9: I cannot follow the definition of "percent end-member limited". If samples were generated from three synthetic end-members with constrains of each within the fraction of 0-1 and all summing to 1, how come are some outside of the triangle? Standard deviation or normalized uncertainties? Also need to explain if the normalization is the same as the one shown in an earlier figure. What do components X and Y mean here? PC components? If so, say so!

**Percent end-member limited is a way we used to demonstrate how much influence those end-member constraints puts on the random generated samples. We modified some of the sentences to make it clear, and we added an additional sentence to the end of the caption that "'Percent end-member limited' is an intuitive alternation of describing the different degrees of sample dispersion from the inner center of the convex-hull.", to clarify the meaning of the term. We also added "Component X and Y represent the two synthetic measures, X [-] and Y [-], in Figure 8." to explain the meaning of X and Y.**

Table 1: Title should be placed above the table (different from figures) unless the journal requires to be placed under. Indicate the number of end-members for each block. What are their fractional contributions? Are the contributions from fourth and fifth end-members significant?

**The table has been updated to include the number of end-members. We did not calculate the contributions for the reasons we have listed above at Major Concerns section.**

Table 2: Title should be above the table if the journal does not require to be under.

**Thank you. We have replaced the table title.**

---

## Author Response (AR4)

**Response to Reviewer**

**A *data-driven method for estimating the composition of end-members from stream water chemistry timeseries* by Esther Xu Fei and Ciaran J. Harman**

Jan. 5th, 2021

We would like to thank the reviewer for their thoughtful comments and efforts towards improving the quality of our manuscript. In the following responses, we have updated the manuscript in four main categories:

1. We addressed the concern regarding the end-member distance by adding two paragraphs, one figure, and one table.
2. We clarified the "percent end-member limited" confusion by adding one paragraph and sentences that referring to this concept.
3. We fixed minor problems accordingly.
4. In addition to reviewer's comments, we also made minor revisions to the methodology section to further clarify how CHEMMA works.

Below our responses are in green, and the feedback from reviewers and editors is in black.

I appreciate authors' patience and continuous effort to improve their work and the manuscript. Compared to its last version, this version has been significantly improved. Publication of this work may have a high impact. Thus, the quality of its presentation must be ensured.

My current concerns focus on the following:
(1) Authors misunderstood the end-member distance, which cannot be inferred from Figure 3. It is not the geometrical distance between two end-members in the mixing diagram, but the distance between U- and S-spaces (defined by Christophersen and Hooper, 1992) for individual solutes of each end-member. It is a measure of fitness an end-member to the mixing space. Examining the end-member distances may provide a strong support of the end-member characterization by CHEMMA. Field sampling from a specific location may not perfectly represent the characteristics of an end-member due to spatial (also temporal at some point) variation, but CHEMMA may do better in this case. Any improvement can be reflected by decreases in end-member distances between U- and S-spaces (the shorter the end-member distances, the better it fits to the mixing space; perfect if zeros for all solutes for an end-member).
This calculation is needed because in this study there is a lack of quantitative evaluation of end-member compositions by CHEMMA, other than qualitative comparison with field measurements.

We would like to thank the reviewer for suggesting calculating the distance between U-space and S-space for each end-member. We added a new section in the methodology section to include the paragraph after L185 as "Assessing the goodness of fit". Then we modified the

previous paragraph from L185-188 and added another paragraph to reflect the quantitative assessment of goodness of fit in the Methodology section:

*"There are several metrics that arise naturally from the CHEMMA framework which could be used to assess the goodness of fit of the inferred mixing subspace. The first and second are the centroid and within-cluster variance of each inferred end-member, which will tend to increase as the number of end-members increases. The third is the orthogonal projection distance from observation space to the mixing subspace, which will be smaller when the end-member lies closer to the linear subspace where the rest of the data live. In this paper, we consider a new cluster to be tenable as a proper end-member if: 1) the spread of previously identified clusters remains similar or decreases, 2) the cluster itself has a reasonable variance, and 3) the orthogonal projection distances of previously identified end-members do not significantly increase after adding a new end-member.*

*We can also assess the degree to which CHEMMA and field-sampled end-members are "similar" to the stream chemical signatures. Field end-member candidate samples typically rely on a few grab samples (for example in \citet{Hooper1990} the groundwater were based on samples from a single well) which may insufficiently sample the overall source variability. CHEMMA end-members may provide a better idea of the time-space averaged chemical signature of a source than the field samples. One way to examine this is to look at the difference between an end-member's composition and its composition when projected into the reduced-rank $k-1$ principal component subspace. This can be done for both field-sampled and CHEMMA end members. A summary measure of that difference is the Euclidean distance of the end-member from the reduced-rank subspace. Where that distance is shorter the end-member has a chemical profile that is aligned with that which is typically found in the stream. This distance can be calculated from the loadings on remaining $n-k+1$ principal components."*

Accordingly, in the result section, we have added one paragraph after L213 to reflect the result from calculating this distance with one figure and one table:

*"The three CHEMMA end-members are also located closer to the subspace spanned by the $k-1$ PC than the original three field-sampled end-members. The orthogonal projection distances are given in Table \ref{tab: dis}, and show that the CHEMMA end-members are more similar to the stream chemistry than the field samples, particularly for the groundwater end-member (field sample distance: 0.814, CHEMMA sample distance: 0.450). The differences in the chemical signatures of the groundwater end-members and their projections in the data subspace are shown in Figure \ref{fig: distance} (with concentrations given in standardized units, left for field samples and right for CHEMMA predictions). The CHEMMA end member's Alkalinity, SO$_4$, Ca values in*

*particular are much closer to that of the data subspace than the field-sampled end-member, which is indicated by the shorter distance from the original 6-D chemical profile in dots (blue for field samples and red for CHEMMA predictions) to the 2-D mixing space profile in flat caps (orange for field samples and green for CHEMMA predictions). Only for Si is the field-sampled value closer. After PCA dimension reduction, both field-sampled profile and CHEMMA-predicted profile are close in the standardized solute space. It is worth noting that CHEMMA does not require dimensional reduction: PCA is only needed to determine the number of end-members."*

We also added the following section at the beginning of "3.2 Dimensionality and DTMM" section to present the rest of the results of Table 3:

*"For 4 CHEMMA end-member case in Table \ref{tab: dis}, the orthogonal projection distances of organic, hillslope, and groundwater end-members decrease/remain similar with 3 CHEMMA end-member case. Adding a fifth end-member significantly increase the projection distance of identified 4-th end-member."*

(2) Thanks for explaining "percent end-member limited"! But I think it is still hard to follow, particularly because it is not explicitly described in the method section. If I understand it right, samples were generated randomly first and then screened by end-member criteria, with those outside the triangle discarded. If so, it should be called "percent of samples limited by end-member criteria", which makes more sense.

More importantly, I do not understand the point you were making using those outside samples. Did you mean the more samples lie outside the triangle the better end-members were characterized, as shown by Figures 7 and 8? If this is indeed what you meant, it does not seem to be correct (well, actually, right results for wrong reasons) and even misleading. It is misleading because readers could consider samples outside the triangle here in your case the same as outliers we have often seen in our real samples. Essentially, cases 4, 5, and 6 did better because they contained more "extreme" samples that are closer to true end-members. I use "extreme" here to indicate samples having extremely high fraction of one end-member but extremely low fraction(s) of the others. This is analogic to using baseflow for groundwater in some real-world cases. If this is what you really meant, I think you have to change your description and make sure readers will not be misled.

Actually, a better, more intuitive way to do this is to generate random samples while applying the constrains at the same time, with varying constrains of end-member fractions, e.g., case 1: 0.4-0.6, case 2: 0.3-0.7, case 3: 0.2-0.8, case 4: 0.1-0.9, and case 5: 0-1.0. This will control the number of samples closer to the true end-members.

We would like to thank the reviewer for pointing out the confusion here. First, we have adjusted several words in sentences from the methodology section to increase the clarity. Second, the 'percent end-member limited' is a measure that is specific to the way we generated

the synthetic data, but serves its purpose in characterizing the spread of the data relative to the mixing space. We have modified the paragraph at line 313 to remove the emphasis on this metric and focus on the insight gained:

> *For the synthetic dataset, the algorithmic uncertainty becomes insignificant when the data cloud just begins to be constrained by the end members. In case 4 in Figure 7) less than 1% of the random samples generated fell outside the mixing space (and were thus discarded). Note that it is the edges, not the verticies, that have affected the shape of the data cloud at this stage. This suggests that the CHEMMA algorithm does not require that there be `extreme' samples containing large contributions from only one end member (i.e. samples close to a vertex in the mixing space). Rather, it can detect mixing structure robustly when the dataset includes samples containing very small contributions of one end member, and intermediate contributions of other (i.e. samples close to an edge/face of the mixing space, but far from a vertex). However, an end-member whose contribution is consistently low may not be effectively detected because it does not affect the shape of the data cloud boundary sufficiently to justify increasing the number of end-members sought (i.e. the number of principal components retained in the analysis plus one).*

(3) Some statements are still pretty speculative (though they may not be incorrect). I suggest to stick with what your data (figures and tables) actually show and avoid stretching too much or being too inclusive. See examples below where this happens.

Thank you for this comment. We have responded to those statements in the section below.

Miscellaneous Comments:
P1/L15-16: This clause is awkward. I think you meant that "a subset of samples with extremely high and low fractions of end-member contributions ... under extreme hydrologic conditions".
P11/L317-319: Both statements are awkward, but I know what you meant. "...small contribution ..." in the first sentence refers to samples having extremely high and low fractions of end-member contributions under extreme hydrologic conditions and the second refers to some potential end-members exist in the catchment but their impact to streamflow and chemistry is insignificant. If so, change them.
P12/L345: I understand it but it would be hard for others to follow or to get it right away. I think you were talking about extreme samples here as I mentioned earlier.

We would like to collectively clarify the meaning of these two sentences. As discussed in our response to (2) above, we mean that to successfully identify end-members CHEMMA requires a subset of samples that are lacking in one particular end-member but are enriched in others. We have modified the last sentence in the abstract to "*The results suggest that the mixing space can be identified robustly when the dataset includes samples that contain extremely small contributions of one end-member -- samples containing extremely large contributions from one end-member are not necessary, but do reduce uncertainty about the end-member composition.*"

For L317-319, see our response to major comment (2) above. For L345, we slightly changed the sentence to "*In other words, the algorithmic uncertainty was essentially eliminated if at least a few samples contained nearly zero contribution from at least one end member.*"

P4-6/L116-146: I think "k" has a different meaning before and after Line 127. I think it means the number of PCs or the number of dimensions before Line 127, but the number of end-members or vertices after Line 127. Because the number of end-member is one more than the number of mixing dimensions, "k" can not be used the same in the two sections.

Thank you for your suggestion. We have corrected the dimension from k to k-1.

P5/L132: Cite "Step 3, Figure 1a" here.

Thank you for your suggestion. We have changed it to "Step 3, from Figure 1a to Figure 1b, notice the changing distribution of the blue points" to reflect the process of projecting samples from the observational space to the principal component space.

P8/L234-235: Good to have such a statement, but it is too casual and lack of specifics.
Thank you for pointing out the need of clarification. We have added a following sentence to clarify it as "CHEMMA identifies sources that can be found through their control on the boundary of the sample space."

P9/L244-245: Why not cite the results of DTMM to support your statement here and also in the conclusion (P12/L359-360)?
P12/L359-360: DTMM results should be cited in the results section.

We have cited the DTMM for L244-245 and add additional sentence to clarify the conflict between DTMM and the rule of one: "An additional dimensionality (additional eigenvector to be retained) can be added until residual structure is unseen or is not improved."

For L359-360, we also added another sentence citing DTMM result to support our conclusion. "DTMM (Hooper 2003) was used to conclude that 1) the dimensionality of Panola dataset is at least 3 (i.e., at least 4 end-members are required), 2) the possible fourth source (end-member) may be weathering products containing calcium and magnesium. CHEMMA was able to identify a fourth end-member with such characteristic without run through DTMM analysis."

P9/L255-256: An example of a speculative statement without a demonstration or support (though not necessarily incorrect).

We have added a citation to Inamdar et al 2013, which corroborates our point.

P11/L330-339: Discussion mode.

We feel that this section works better in the Conclusion and would prefer to keep it here.

P11/L336-338: Awkward and hard to follow.

We have clarified this sentence by changing 'biased' to 'uncertain'

P12/L363-364: Clueless to readers with a sudden introduction of these terms. Citing a perception is better.

We have added a citation where readers can get more information.

P12/L366 (beginning): Awkward and hard to follow.

Changed 'observations' to 'samples'.

P2/L30-33: Cite literature where the statements came from. I think some languages/phrases are not accurate, e.g., "approximately" and "additional end-members" used in the statements.
P2/L46: Try not to use ambiguous phrases. Change to "a similar approach".
P3/L85-86: Not completely true. People used baseflow for groundwater in many cases. Change the statement to "there is not a method ..., other than using baseflow to characterize groundwater ...". Add a reference (e.g., Liu et al., Ecohydrology, 2008).
P8/L214: You mean "four magenta diamonds"?
P8/L229: Not assumed. Using "suggested" may be a better word choice.
P10/L280-286: Parenthesize these sequential numbers to avoid unnecessary confusion with counting numbers.
P11/L342: Fix the grammar here.
P12/L351-354: Use consistent numbering. Used numbers before but letters here.
P12/L361: Change "might" to "should".
P12/L362-369: Use consistent numbering, including parentheses (half or complete) throughout the manuscript.

We would like to thank the reviewer for pointing out these minor problems. We have corrected them accordingly.